# Host and viral determinants of airborne transmission of SARS-CoV-2 in the Syrian hamster

Julia R Port[1]*, Dylan H Morris[2], Jade C Riopelle[1], Claude Kwe Yinda[1], Victoria A Avanzato[1], Myndi G Holbrook[1], Trenton Bushmaker[1], Jonathan E Schulz[1], Taylor A Saturday[1], Kent Barbian[3], Colin A Russell[4], Rose Perry-Gottschalk[5], Carl Shaia[6], Craig Martens[3], James O Lloyd-Smith[2], Robert J Fischer[1], Vincent J Munster[1]*

[1]Laboratory of Virology, Division of Intramural Research, National Institute of Allergy and Infectious Diseases, National Institutes of Health, Hamilton, United States; [2]Department of Ecology and Evolutionary Biology, University of California, Los Angeles, Los Angeles, United States; [3]Rocky Mountain Research and Technologies Branch, Division of Intramural Research, National Institute of Allergy and Infectious Diseases, National Institutes of Health, Hamilton, United States; [4]Department of Medical Microbiology | Amsterdam University Medical Center, University of Amsterdam, Amsterdam, Netherlands; [5]Rocky Mountain Visual and Medical Arts Unit, Research Technologies Branch, Division of Intramural Research, National Institute of Allergy and Infectious Diseases, National Institutes of Health, Hamilton, United States; [6]Rocky Mountain Veterinary Branch, Division of Intramural Research, National Institute of Allergy and Infectious Diseases, National Institutes of Health, Hamilton, United States

*For correspondence:
julia.port@nih.gov (JRP);
vincent.munster@nih.gov (VJM)

Competing interest: The authors declare that no competing interests exist.

**Abstract** It remains poorly understood how SARS-CoV-2 infection influences the physiological host factors important for aerosol transmission. We assessed breathing pattern, exhaled droplets, and infectious virus after infection with Alpha and Delta variants of concern (VOC) in the Syrian hamster. Both VOCs displayed a confined window of detectable airborne virus (24–48 hr), shorter than compared to oropharyngeal swabs. The loss of airborne shedding was linked to airway constriction resulting in a decrease of fine aerosols (1–10 μm) produced, which are suspected to be the major driver of airborne transmission. Male sex was associated with increased viral replication and virus shedding in the air. Next, we compared the transmission efficiency of both variants and found no significant differences. Transmission efficiency varied mostly among donors, 0–100% (including a superspreading event), and aerosol transmission over multiple chain links was representative of natural heterogeneity of exposure dose and downstream viral kinetics. Co-infection with VOCs only occurred when both viruses were shed by the same donor during an increased exposure timeframe (24–48 hr). This highlights that assessment of host and virus factors resulting in a differential exhaled particle profile is critical for understanding airborne transmission.

## eLife assessment

This manuscript describes rigorous experiments that provide a wealth of virologic, respiratory physiology, and particle aerodynamic data pertaining to aerosol transmission of SARS-CoV-2 between infected Syrian hamsters. The significance of the paper is **fundamental** because infection is compared between alpha and delta variants, and because viral load is assessed via numerous assays

(gRNA, sgRNA, TCID) and in tissues as well as the ambient environment of the cage. The strength of evidence is **compelling**.

## Introduction

Transmission by aerosolized virus particles has been a major contributor to the spread of SARS-CoV-2 (*Zhang et al., 2020*; *Port et al., 2022*; *Boone and Gerba, 2007*; *Goldman, 2020*; *CDC, 2021a*; *Pitol and Julian, 2021*). Although highly efficient in preventing severe disease, vaccines do not significantly reduce transmission of variants of concern (VOCs) (*CDC, 2021b*). Transmission occurs when people release respiratory droplets carrying virus during (e.g.) speaking, singing, breathing, sneezing, or coughing. Droplet size and half-life in the air are not uniform (*Morawska et al., 2009*; *Stadnytskyi et al., 2020*) and depend on speech and breathing patterns (*Johnson and Morawska, 2009*), COVID-19 severity, and physiological parameters such as age (*Coleman et al., 2022*; *Edwards et al., 2021*). As with influenza (*Milton et al., 2013*), SARS-CoV-2 RNA was detectable mostly in fine aerosols in humans, as opposed to coarse aerosols (*Coleman et al., 2022*). It is not clear how exhaled droplet size, breathing patterns and even the quantity of exhaled infectious virus itself fundamentally contribute to the airborne transmission efficiency in vivo and how COVID-19 directly influences additional physiological factors which may contribute to fine aerosol production. There is reportedly large heterogeneity in the transmission potential of individuals. Superspreading events have been reported numerous times throughout the pandemic and are suggested to be a major driver (*Sun et al., 2021*; *Yang et al., 2021*). They are thought to arise from a combination of biological, social, and chance factors. While human epidemiology and modeling studies have highlighted various factors which may contribute to SARS-CoV-2 transmission heterogeneity, including viral load (*Goyal et al., 2021*), much of the observed variance remains poorly understood. These factors are currently best studied in small animal models like the Syrian hamster, which allow for stringent and controlled experimental comparisons. The Syrian hamster model has been widely used to study SARS-CoV-2 transmission *Muñoz-Fontela et al., 2020*; it recapitulates human contact, fomite and, importantly, airborne short distance and fine aerosol transmission (*Port et al., 2022*; *Sia et al., 2020*; *Rosenke et al., 2020*; *Port et al., 2021b*). In this model, highest efficiency of short-distance airborne transmission was observed before onset of weight loss and acute lung pathology, peaking at 1 day post inoculation and correlating to the highest virus loads in the upper respiratory tract of donor animals (*Ganti et al., 2022*). Data on lung function loss in the Syrian hamster model after SARS-CoV-2 infection is available (*Port et al., 2021a*; *Halfmann et al., 2022*), and virus has been demonstrated in exhaled droplets (*Hawks et al., 2021*). Yet, a systematic study that addresses how airborne transmission potential depends on these features, along with recognized influences of sex and VOC, has not been performed. The study of these contributing factors would allow us to address how they come together to shape transmission outcomes.

Here, we introduce a mathematical model delineating for Alpha and Delta VOCs the relationship between exhaled infectious virus and virus detected in the upper respiratory tract during infection and longitudinally detail the changes in lung function, respiratory capacity, and exhaled particle profiles. Finally, we assess the airborne transmission competitiveness and heterogeneity in vivo of Alpha and Delta.

## Results

### Peak infectious SARS-CoV-2 in air samples is detected between 24 hr and 48 hr post infection

Structural modeling and pseudotype-entry comparison suggested that the Syrian hamster model should recapitulate the entry-specific competitive advantage of Delta over Alpha observed in humans (*Figure 1—figure supplement 1A–D*). Syrian hamsters were inoculated with a low dose ($10^3$ TCID$_{50}$, intranasal [IN], N=10 per group) of SARS-CoV-2 Delta or Alpha. Animals were monitored for 14 days post inoculation (DPI). We observed no significant differences in weight loss or viral titers in lung or nasal turbinates between the variants (*Figure 1—figure supplement 2A–C*). At 14 DPI, hamsters (N=5) mounted a robust anti-spike IgG antibody response, and the overall binding pattern was similar

between Alpha and Delta (*Figure 1—figure supplement 2D and E*). In a live virus neutralization assay, homologous virus was neutralized significantly better as compared to the heterologous variant *Figure 1—figure supplement 2F and G*), but no significant difference was determined between the neutralization capacity against the respective homologous variant (median reciprocal virus neutralization titer = 320 (Alpha anti-Alpha)/ 320 (Delta anti-Delta), p=0.9568, N=5, ordinary two-way ANOVA, followed by Tukey's multiple comparisons test.

We determined the window of SARS-CoV-2 shedding for Alpha and Delta using swabs from the upper respiratory tract and air sampling from cages, quantifying virus using gRNA, sgRNA, and infectious virus titers. Oral swabs remained positive for gRNA and sgRNA until 7 DPI, but infectious virus dropped to undetectable levels after 4 DPI in most individuals (*Figure 1—figure supplement 3A*). Cage air was sampled during the first 5 days of infection in 24 hr time windows from cages containing two or three animals, grouped by sex. gRNA and sgRNA were detectable as early as 1 DPI in 50% of air samples and remained high through 5 DPI, while infectious virus peaked at on 2 DPI and was detectable for a shorter window, from 1 to 4 DPI (*Figure 1—figure supplement 3B*).

## Mathematical modeling demonstrates airborne shedding peaks later and declines faster than oral swab viral load

We quantified heterogeneity in shedding by variant, sex, and sampling method by fitting a mathematical model of within-hamster virus kinetics (see Appendix) to the data. This served to correlate parameters which are easier to measure, such as RNA in the oral cavity, to the quantity of greatest interest for understanding transmission (i.e. infectious virus in the air per unit time). To do this, we jointly inferred the kinetics of shed airborne virus and parameters relating observable quantities (e.g. plaques from purified air sample filters) to the actual longitudinal shedding. The inferential model uses mechanistic descriptions of deposition of infectious virus into the air, uptake from the air, and loss of infectious virus in the environment to extract estimates of the key parameters describing viral kinetics, as well as the resultant airborne shedding, for each animal. Virus was detectable and peaked earlier in oral swabs (approximately 24 hr post inoculation) than virus sampled from the air (approximately 48 hr post inoculation), and quantity of detected virus declined slower in the swabs (*Figure 1A and B*). gRNA and sgRNA declined slower than infectious virus both in the air and in swabs. Oral swab data was an imperfect proxy for airborne shedding, even when we directly quantified infectious virus titers. This was due to a lag between peak swab shedding and peak airborne shedding. Inferred within-host exponential growth and decay rates were similar for the two variants. For both variants, males shed more virus than females, even after accounting for males' higher respiration rates in measurements of shedding into the air. We found a slightly higher ratio of infectious virus to sgRNA in air samples for Delta than for Alpha (*Figure 1B*, *Figure 1—figure supplement 3C and D*). We also found substantial individual-level heterogeneity in airborne shedding, even after accounting for sex and variant (*Figure 1B*). For example, air samples from cage 5 had more than twice as many peak plaques per capita than cage 6, even though both cages contained hamsters of the same sex, inoculated by the same dose, route, and variant. Our model captures this, with substantial inferred heterogeneity in individual airborne shedding in PFU per h, both in timing and in height of peak (*Figure 1B*).

## Changes in breathing profile after SARS-CoV-2 infection precede onset of weight loss and are variant and sex-dependent

Pathology in nasal turbinates and lungs did not differ significantly between animals (*Figure 2—figure supplement 1*). Pathological changes were consistent with those described previously for COVID-19 in Syrian hamsters after intranasal inoculation with other SARS-CoV-2 strains (*Rosenke et al., 2020*). Whole body plethysmography (WBP) was performed. We focused the analysis on the first 5 days after inoculation, in which changes in virus shedding and release into the air were observed (*Figure 1— figure supplement 2A*). Expiratory time (Te), inspiratory time (Ti), percentage of breath occupied by the transition from inspiration to expiration (TB), end expiratory pause (EEP), breathing frequency (f), peak inspiratory flow (PIFb), peak expiratory flow (PEFb), tidal volume (TVb), minute volume (MVb), and enhanced pause (Penh) were used to assess changes in pulmonary function throughout infection. Principal component analysis was used to determine trends in lung function changes across all groups (*Figure 2A*). This revealed a large degree of inherent variation in individual hamster plethysmography measures. Before inoculation there was no discernible pattern to the clustering observed besides

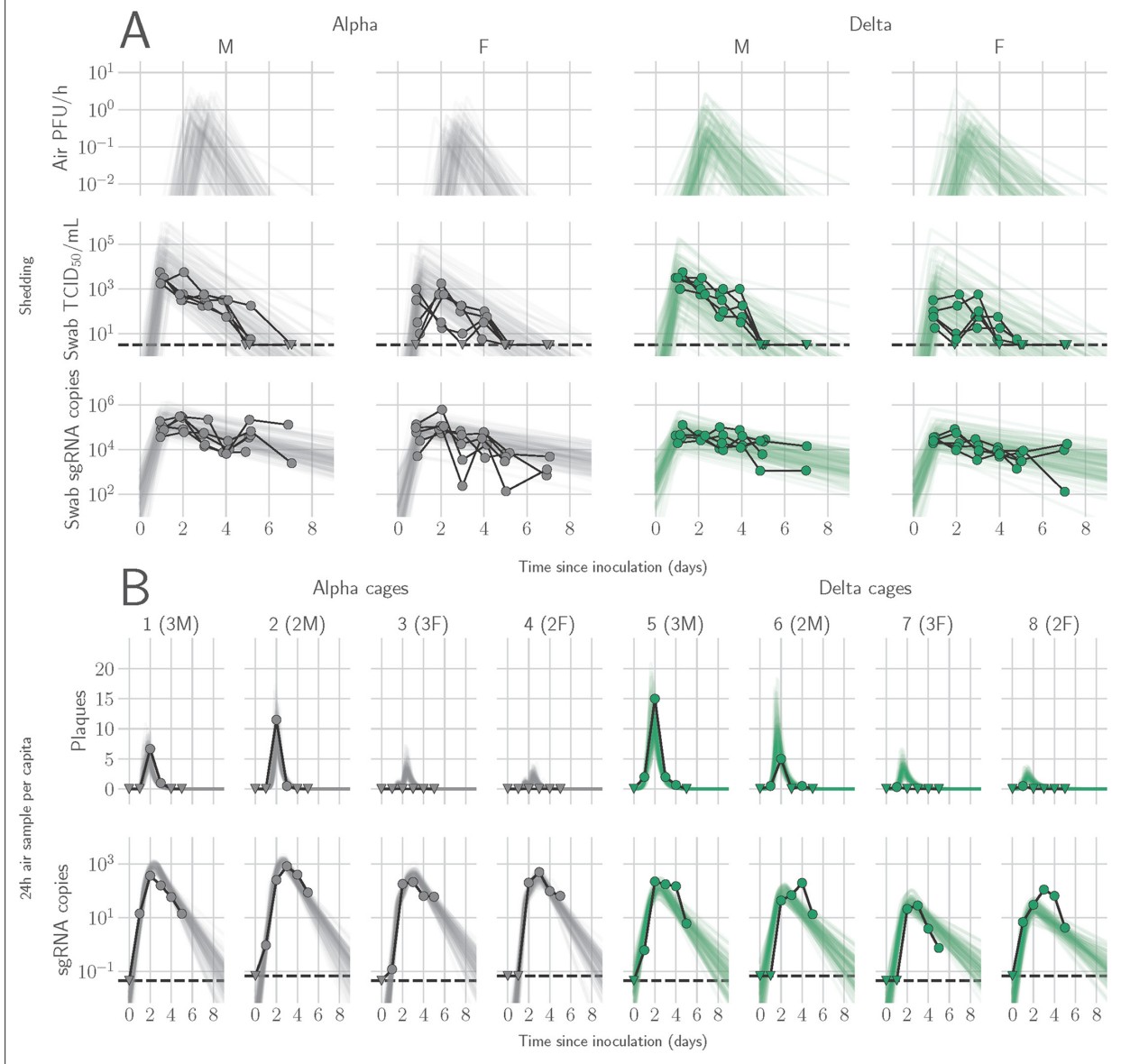

**Figure 1.** Alpha and Delta variant shedding profiles in oral swabs and air samples. Syrian hamsters were inoculated with $10^3$ TCID$_{50}$ via the intranasal route with Alpha or Delta. (**A**) Comparison of swab viral load and virus shedding into the air. Inferred profile of air shedding in PFU/h compared to sgRNA levels and infectious virus titers (TCID$_{50}$/mL) in oropharyngeal swabs collected 1, 2, 3, 4, 5, and 7 DPI. Semitransparent lines are 100 random draws from the inferred posterior distribution of hamster within-host kinetics for each of the metrics. Joined points are individual measured timeseries for experimentally infected hamsters; each set of joined points is one individual. Measurements and inferences shown grouped by variant and animal sex. Measurement points are randomly jittered slightly along the x (time) axis to avoid overplotting. (**B**). Viral sgRNA and infectious virus (PFU) recovered from cage air sample filters over a 24 hr period starting at 0, 1, 2, 3, 4, and 5 DPI. Points are measured values, normalized by the number of hamsters in the cage (2 or 3) to give per-capita values. Downward-pointing arrows represent virus below the limit of detection (0 observed plaques or estimated copy number corresponding to Ct ≥40). Semitransparent lines are posterior predictions for the sample that would have been collected if sampling started at that timepoint; these reflect the inferred underlying concentrations of sgRNA and infectious virus in the cage air at each timepoint and are calculated from the inferred infection kinetics for each of the hamsters housed within the cage. 100 random posterior draws shown for each cage. Cages housed 2 or 3 hamsters; all hamsters within a cage were of the same sex and infected with the same variant. Column titles show cage number and variant, with number of and sex of individuals in parentheses. Dotted lines limit of detection. Grey = Alpha, teal = Delta, p-values are indicated where significant. Abbreviations: sg, subgenomic; TCID, Tissue Culture Infectious Dose; PFU, plaque forming unit; F, female; M, male; DPI, days post inoculation.

The online version of this article includes the following figure supplement(s) for figure 1:

**Figure supplement 1.** Alpha and Delta variant spike interaction with hamster ACE2.

*Figure 1 continued on next page*

**Figure supplement 2.** Schematic overview of inoculation experiment.

**Figure supplement 3.** Window of Alpha and Delta variant shedding profiles.

a slight separation by sex. Beginning at 2 DPI, we observed a separation of infected and control animals. This coincided with the observation that all SARS-CoV-2 animals visibly decreased activity levels after 2 DPI, reducing exploratory activity and grooming with sporadic short convulsions which may represent coughing. No single parameter had an overwhelming influence on clustering, though several parameters contributed strongly across all days: Te, Ti, TB, EEP, f, PIFb, PEFb, TVb, and MVb (*Figure 2B and C*).

Broad patterns emerged by variant and by sex. Cumulative Penh AUC values for all infected groups were increased compared to the sex-matched control hamsters (p=0.022, Kruskal-Wallis test, N=4 for Alpha and Delta, N=5 for controls). The median Penh AUC values for Alpha, Delta, and control males were 0.741, 2.666, and 0.163, respectively (p=0.062). The median Penh AUC values for Alpha females, Delta females, and control females were 1.783, 2.255, and 0.159, respectively (p=0.019). At 4 DPI, the median fold change Penh values for Alpha males and Delta males were 0.793 and 1.929, respectively, as compared to 0.857 for control males. The corresponding Penh values for Alpha, Delta, and control females were 1.736, 1.410, and 1.008, respectively. The separation on 4 DPI did not translate to significant changes in more traditional measures of respiratory function, including f, TVb, and MVb.

## Changes in exhaled aerosol aerodynamic profile after SARS-CoV-2 infection precede acute disease, are variant and sex-dependent

Alpha and Delta inoculated groups (N=10 each) and a control group (N=10) were individually evaluated on 0, 1, 3, and 5 DPI. To normalize the particle counts between animals we focused on the percentage of particles in each size range. Across each variant group, particle diameter size <0.53 µm was the most abundant (*Figure 3A*). No consistent, significant overall change in the number of overall particles across all sizes was observed between groups (*Figure 3—figure supplement 1C*). Particles between 1 and 10 µm in diameter, most relevant for fine aerosol transmission (*Wang et al., 2021*), were examined. At baseline (0 DPI), females across all groups produced a higher proportion of droplets in the 1–10 µm diameter range compared to males (*Figure 3A*). At 3 DPI, the particle profiles shifted toward smaller aerodynamic diameters in the infected groups. At 5 DPI, even control animals demonstrated reduced exploratory behavior, resulting in a reduction of particles in the 1–10 µm range, which could be due to acclimatization to the chamber. This resulted in an overall shift in particle size from the 1–10 µm range to the <0.53 µm range. To analyze these data, individual slopes for each animal were calculated using simple linear regression across the four timepoints (Percent ~Intercept + SlopeDay) for percent of particles in the <0.53 µm range and percent of particles in the 1–10 µm range and multiple linear regression was performed (*Figure 3B*). Females had a steeper decline at an average rate of 2.2 per day after inoculation in the percent of 1–10 µm particles (and a steeper incline for <0.53 µm) when compared to males, while holding variant group constant. When we compared variant group while holding sex constant, we found that the Delta group had a steeper decline at an average rate of 5.6 per day in the percent of 1–10 µm particles (and a steeper incline for <0.53 µm); a similar trend, but not as steep, was observed for the Alpha group.

The estimated difference in slopes for Delta vs. controls and Alpha vs. controls in the percent of <0.53 µm particles was 5.4 (two-sided adjusted p=0.0001) and 2.4 (two-sided adjusted p=0.0874), respectively. The estimated difference in slopes for percent of 1–10 µm particles was not as pronounced, but similar trends were observed for Delta and Alpha. Additionally, a linear mixed model was considered and produced virtually the same results as the simpler analysis described above; the corresponding linear mixed model estimates were the same and standard errors were similar.

## Alpha and Delta VOC attack rates reveal minimal individual risk of dual infection in vivo

We next compared attack rates between Alpha and Delta during a 4 hr exposure window at 200 cm distance. Groups of sentinels (N=4 or 5) were exposed to two donor animals, one inoculated with Alpha and one inoculated with Delta (*Figure 4A*).

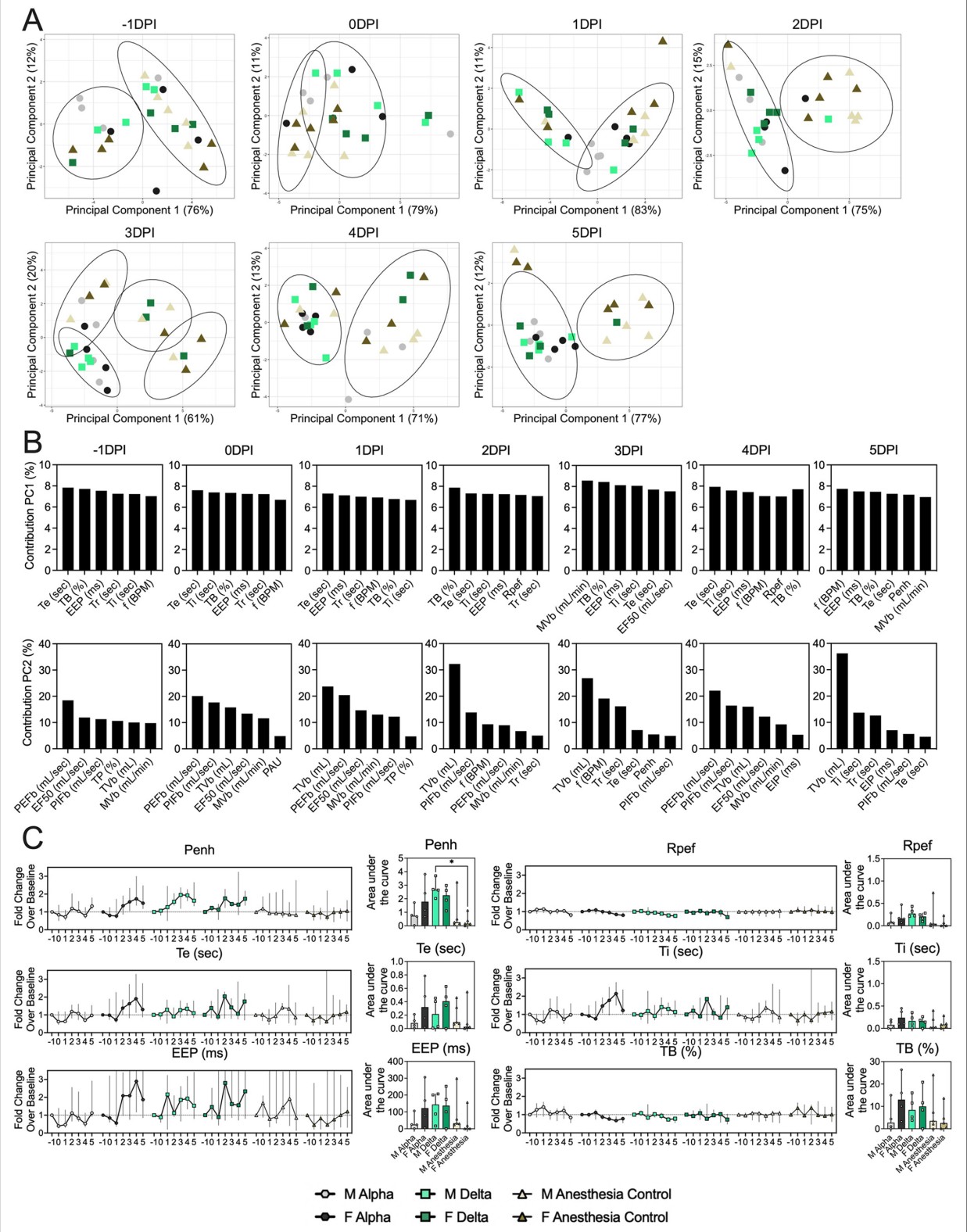

**Figure 2.** Lung function and breathing changes after SARS-CoV-2 infection with Alpha and Delta. Syrian hamsters were inoculated with $10^3$ TCID$_{50}$ via the intranasal route with Alpha or Delta. (**A**) Lung function was assessed on days −1, 0, 1, 2, 3, 4, and 5 by whole body plethysmography. Principal component analysis was used to investigate individual variance. Depicted are principal component (PC) 1 and 2 for each day, showing individual animals (colors refer to legend on right, sex-separated) and clusters (black ellipses). (**B**) Individual loading plots for contributions of top 6 variables to PC1 and

*Figure 2 continued on next page*

*Figure 2 continued*

2 at each day. (**C**) Relevant subset of lung function parameters. Line graphs depicting median and 95% CI fold change values (left) and area under the curve (AUC, right), Kruskal-Wallis test, p-values indicated where significant. Grey = Alpha, teal = Delta, beige = anesthesia control, light = male, dark = female. Abbreviations: Expiratory time (Te), inspiratory time (Ti), percentage of breath occupied by the transition from inspiration to expiration (TB), end expiratory pause (EEP), breathing frequency (f), peak inspiratory flow (PIFb), peak expiratory flow (PEFb), tidal volume (TVb), minute volume (MVb), enhanced pause (Penh), male (M), female (F).

The online version of this article includes the following figure supplement(s) for figure 2:

**Figure supplement 1.** Respiratory tract pathology after SARS-CoV-2 infection with Alpha and Delta.

sgRNA in oral swabs taken on 1 DPI varied between animals (*Figure 4B*). Sentinels were either exposed first for 2 hr to one variant and then for 2 hr to the second (*Figure 4C*, first 4 iterations), or to both variants at the same time for 4 hr (last three iterations). Transmission was confirmed by sgRNA in oral swabs collected from all sentinels at 2, 3, and 5 DPE. On 2 DPE, N=13/34 sentinels were positive for sgRNA in oral swabs, N=19/34 on 3 DPE and N=27/34 on 5 DPE. Swabs from 3 DPE and 5 DPE were sequenced, and the percentage of reads mapped to Alpha, and Delta were compared (*Figure 4D*).

All animals had only one variant detectable on day 3. In total, 12 sentinels were infected with Alpha and 7 with Delta by 3 DPE. At 5 DPE, slightly more sentinels shed Alpha (cartoon hamster representation in *Figure 4D* depicts majority variant for each individual across both sampling days; *Supplementary file 1* lists sequencing results). Interestingly, we observed one superspreading event in iteration A, in which one donor animal transmitted Alpha to all sentinels. For all other iterations, either both donors managed to transmit to at least one sentinel, or not all sentinels were infected. For the iterations with simultaneous exposure, attack rates were similar and statistically indistinguishable: Alpha = 50 %, Delta = 42.8%. In one simultaneous exposure (iteration F), three sentinels had both Delta and Alpha detectable at 5 DPE. In two, Delta was dominant, and in one Alpha, always with the other variant in the clear minority (<15%). We did not observe any other such coinfections (defined as a PCR-positive animal with both Alpha and Delta at 5% frequency or higher by NGS). This led us to ask whether there was virus interference in sequential exposures - that is, whether established infection with one variant could reduce the probability of successful infection given a later exposure.

To assess this, we used our within-host dynamics model to calculate the estimated infection probabilities for Alpha and Delta for each sentinel in each iteration, assuming each sentinel is exposed independently, but accounting for the different exposure durations, donor sexes, and donor viral load (as measured by oral swabs). From those probabilities, we then calculated posterior probability distributions for the number of co-infections predicted to occur in each iteration if Alpha and Delta infections occurred independently and did not interfere with each other (*Appendix 1—figures 2–4*). We found that our observed coinfections were consistent with this null model; our data do not provide clear evidence of virus interference during sequential exposure, though they also do not rule out such an effect. No difference in virus replication or disease severity was observed between the sentinels infected with Alpha or Delta (*Figure 4—figure supplement 1*).

## Limited sustainability of heterologous VOC populations through multiple rounds of airborne transmission

To assess the transmission efficiency in direct competition between the Alpha and Delta VOCs, we conducted an airborne transmission experiment over three subsequent rounds of exposure (*Figure 5A*). Donor animals (N=8) were inoculated IN with $5 \times 10^2$ TCID$_{50}$ of Alpha and $5 \times 10^2$ TCID$_{50}$ Delta (1:1 mixture) and eight sentinels were exposed (Sentinels 1, 1:1 ratio) on 1 DPI for 24 hr (first chain link, exposure window: 24–48 hr post inoculation of the donors) (*Figure 5*). Two days after the start of this exposure, the eight sentinels were placed into the donor side of a new cage and eight new sentinels (Sentinels 2) were exposed for 24 hr (second chain link, exposure window 48–72 hr post exposure start of the Sentinels 1). Again, 2 days after exposure start, this sequence was repeated for Sentinels 3 (third chain link, exposure window 48–72 hr post exposure start of the Sentinels 2). All animals were individually housed between exposures, and after exposure as well for the sentinels. We assessed viral presence in oropharyngeal swabs taken from all animals at 2 and 5 DPI/DPE. While all Sentinels 1 demonstrated active shedding at 2 and 5 DPE, in the Sentinels 2 group no viral RNA was detected in 2/8 animals and no infectious virus in 4/8 by 5 DPE. In the Sentinels 3 group, sgRNA

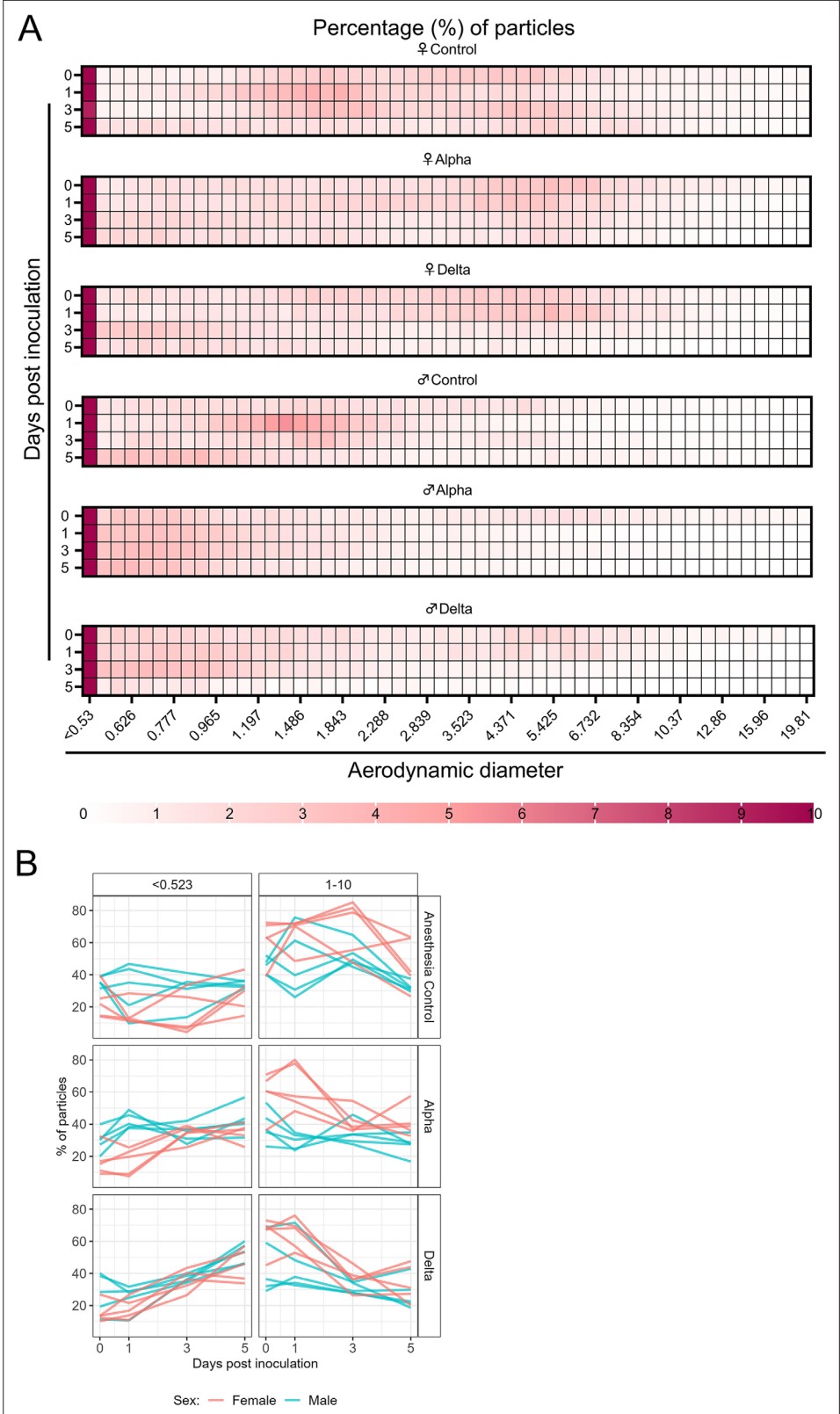

**Figure 3.** Aerodynamic particle analysis of SARS-CoV-2 infected hamsters. (**A**) Syrian hamsters were inoculated with $10^3$ TCID$_{50}$ via the intranasal route with Alpha or Delta. Aerodynamic diameter profile of exhaled particles was analyzed on days 0, 1, 3, and 5. Heatmap shows rounded median percent of total particles across groups, including the anesthesia control group (N=10, comprising five males and five5 females). Colors refer to scale

*Figure 3 continued on next page*

*Figure 3 continued*

below. (**B**) For each animal, line graphs of the percent of particles in the <0.53 and 1–10 μm diameter range by variant group and sex indicated by color. Multiple linear regression performed for each diameter range with group and sex as predictors, F-statistic (**Boone and Gerba, 2007**; **Jones et al., 2021**)=9.47 for<0.53 μm model and F-statistic (**Boone and Gerba, 2007**; **Jones et al., 2021**)=2.62 for 1–10 μm model, with Tukey multiple comparison adjustment for the three variant-group comparisons (95% family-wise confidence level). For <0.53 range, Male-Female (estimate = –1.7, standard error = 0.888, two-sided p=0.0659); Alpha-Control (estimate = 2.41, standard error = 1.09, two-sided p=0.0874), Delta-Control (estimate = 5.40, standard error = 1.09, two-sided p=0.0001), Delta-Alpha (estimate = 2.99, standard error = 1.09, two-sided p=0.0280). For 1–10 range, Male-Female (estimate = 2.19, standard error = 1.23, two-sided p=0.0875); Alpha-Control (estimate = –0.633, standard error = 1.51, two-sided p=0.9079), Delta-Control (estimate = –3.098, standard error = 1.51, two-sided p=0.1197), Delta-Alpha (estimate = –2.465, standard error = 1.51, two-sided p=0.2498). Grey = Alpha, teal = Delta, beige = anesthesia control, red = female, blue = male.

The online version of this article includes the following figure supplement(s) for figure 3:

**Figure supplement 1.** Exhaled particle profiles of Syrian hamsters.

and infectious virus were only detected robustly in one animal on 5 DPE. In contrast to donor animals, all infected sentinels exhibited higher shedding on day 5 compared to day 2 2 DPI / 5 DPI Donors: median gRNA = 7.8 / 6.9 copies/mL (Log$^{10}$), median sgRNA = 7.2 / 6.4 copies/mL (Log$_{10}$), median infectious virus titer = 2.3 / 0.5 TCID$_{50}$/mL (Log$_{10}$); Sentinels 1 (median gRNA = 7.2 / 7.4 copies/mL (Log$_{10}$), median sgRNA = 6.4 / 6.9 copies/mL (Log$_{10}$), median infectious virus titer = 2.9 / 2.6 TCID$_{50}$/mL (Log$_{10}$); Sentinels 2=median gRNA = 3.7 / 5.4 copies/mL (Log$_{10}$), median sgRNA = 1.8 / 3.0 copies/mL (Log$_{10}$), median infectious virus titer = 0.5 / 1.6 TCID$_{50}$/mL (Log$_{10}$)) (**Figure 5B**). Taken together, this evidence suggests that the infectious shedding profile shifts later and decreases in magnitude with successive generations of transmission. This could be explained by lower exposure doses causing lower and slower infections in the recipients.

We then proceeded to compare the viral loads in the lungs and nasal turbinates at 5 DPE. Viral gRNA was detected in the lungs (**Figure 5C**) and nasal turbinates (**Figure 5D**) of all Donors (lungs: median gRNA = 9.7 copies/gr tissue (Log$_{10}$), nasal turbinates: median gRNA = 6.2 copies/gr tissue (Log$_{10}$)). Interestingly, while the gRNA amount was similar in lungs between Donors and Sentinels 1 (lungs: median gRNA = 9.5 copies/gr tissue (Log$_{10}$)), it was increased in nasal turbinates for the Sentinel 1 group (nasal turbinates: median gRNA = 8.6 copies/gr tissue (Log$_{10}$)). Similarly, sgRNA was increased in Sentinels 1 as compared to Donors in nasal turbinates, but not lungs (Donors = lungs: median sgRNA = 9.4 copies/gr tissue (Log$_{10}$), nasal turbinates: median sgRNA = 5.7 copies/gr tissue (Log10); Sentinels 1=lungs: median sgRNA = 9.2 copies/gr tissue (Log$_{10}$), nasal turbinates: median sgRNA = 8.4 copies/gr tissue (Log$_{10}$)). Viral gRNA above the level of quantification was detectable in 6/8 Sentinels 2 in both lungs and nasal turbinates, yet sgRNA was only detected in 4/8 Sentinels 2 in lungs and 5/8 in nasal turbinates. Even though gRNA was detected in 3/8 Sentinels 3, no animal had detectable sgRNA in either lungs or nasal turbinates, signaling a lack of active virus replication. To confirm this, infectious virus was analyzed in both tissues for the Donors, Sentinels 1, and Sentinels 2 groups (**Figure 5E**). In both tissues titers were marginally higher in Sentinels 1 (median TCID$_{50}$ /gr tissue (Log$_{10}$) Donors: lungs = 8.6, nasal turbinates = 8.0; Sentinels 1: lungs = 8.9, nasal turbinates = 8.8). Infectious virus was present in 6/8 Sentinels 2 in lungs and 5/8 in nasal turbinates.

Hence, even though the exposure interval for the second and third chain links were started 48 hr after the start of their own exposure, not all Sentinels 2 became infected, and only one Sentinel 3 animal became infected and demonstrated shedding. We conducted a separate experiment to assess viral loads in the respiratory tract after SARS-CoV-2 airborne transmission at 2 DPI/DPE. While infectious virus was present in oral swabs from all sentinels, virus in lungs and nasal turbinates was not present in all animals (**Figure 5—figure supplement 1**).

To determine the competitiveness of the variants, we analyzed the relative composition of the two viruses using next generation sequencing (**Figure 5F and G**). Neither variant significantly outcompeted the other. We first compared the percentage of Delta in oral swabs taken on 2 DPI/DPE, the day of exposure of the next chain link. In Donors, no variant was more prevalent across animals or clearly outcompeted the other within one host (median = 56.5% Delta, range = 40.3–69%). After the first transmission event, Delta outcompeted Alpha at 2 DPE (median = 87.3% Delta, range = 19–92.7%),

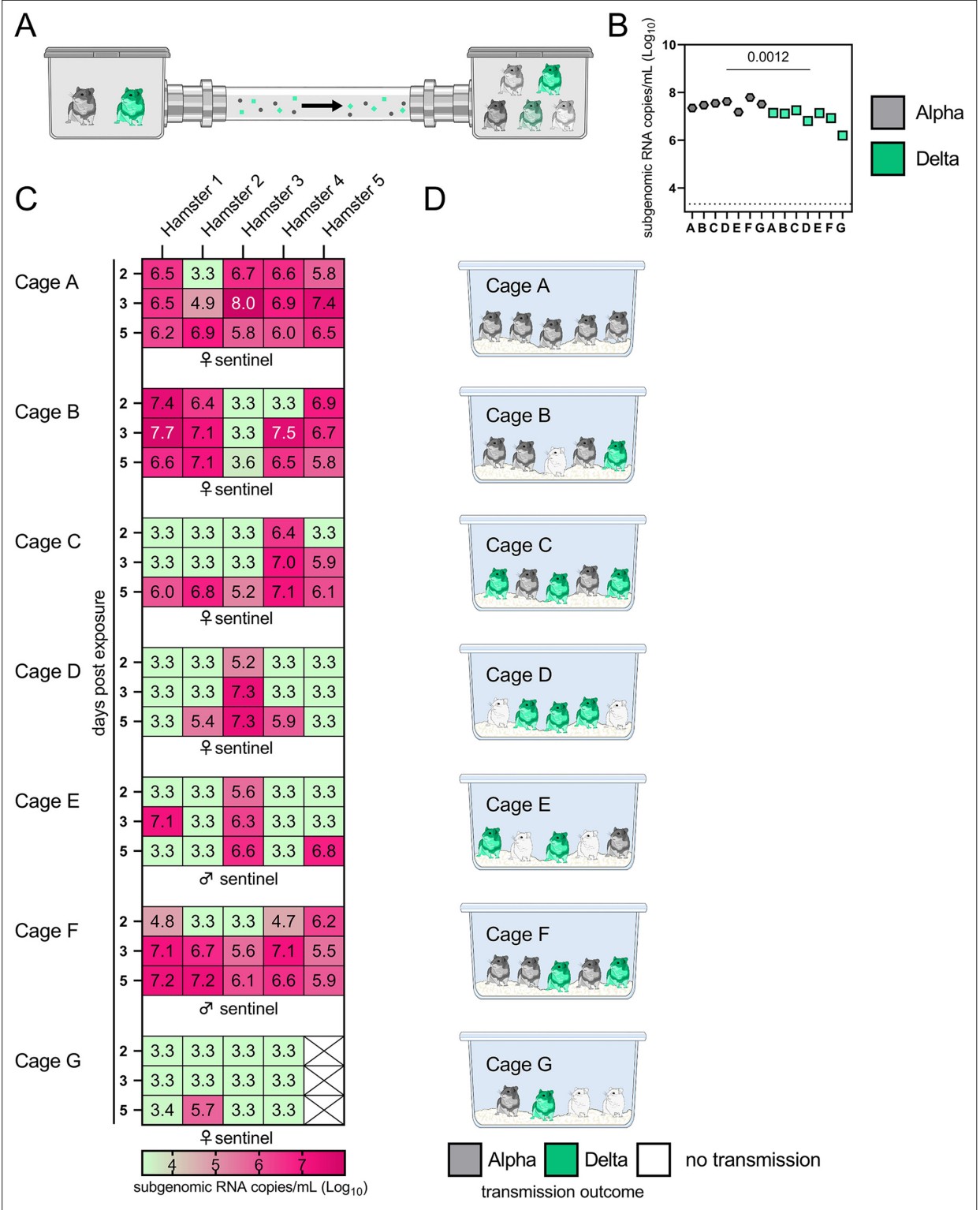

**Figure 4.** Airborne attack rate of Alpha and Delta SARS-CoV-2 variants. Donor animals (N=7) were inoculated with either the Alpha or Delta variant with $10^3$ $TCID_{50}$ via the intranasal route and paired together randomly (1:1 ratio) in 7 attack rate scenarios (**A–G**). To each pair of donors, 1 day after inoculation, 4–5 sentinels were exposed for a duration of 4 hr (i.e. h 24–28 post inoculation) in an aerosol transmission set-up at 200 cm distance. (**A**) Schematic figure of the transmission set-up. (**B**). Day 1 sgRNA detected in oral swabs taken from each donor after exposure ended. Individuals are depicted. Wilcoxon test, N=7. Grey = Alpha, teal = Delta inoculated donors. (**C**) Respiratory shedding measured by viral load in oropharyngeal swabs; measured by sgRNA on days 2, 3, and 5 for each sentinel. Animals are grouped by scenario. Colors refer to legend below. 3.3=limit of detection of

*Figure 4 continued on next page*

*Figure 4 continued*

RNA (<10 copies/rxn). (**D**) Schematic representation of majority variant for each sentinel as assessed by percentage of Alpha and Delta detected in oropharyngeal swabs taken at day 2 and day 5 post exposure by deep sequencing. Grey = Alpha, teal = Delta, white = no transmission.

The online version of this article includes the following figure supplement(s) for figure 4:

**Figure supplement 1.** Airborne competitiveness of Alpha and Delta SARS-CoV-2 variants.

while after the second transmission event, half (N=2/4) the animals shed either >80% either Alpha or Delta. Notably, and in strong contrast to the dual donor experiments described above, every sentinel animal exhibited a mixed infection at 2 DPE, often with proportions resembling those in the donor.

Next, we looked at the selective pressure within the host. By 5 DPI/DPE, no clear difference was observed in Donors (median = 60% Delta, range = 34.3–67.7%), but in the Sentinels 1 group Alpha overtook Delta in three animals (median = 68.3% Delta, range = 17–92.3%), while the reverse was never seen. In one animal, we observed a balanced infection established between both variants at 5 DPE (Sentinel 1.8). In the Sentinels 2 group, Alpha was the dominant variant in N=3/8 animals, and Delta dominated in 3/8 (median = 55% Delta, range = 17–92.7%). The one Sentinel 3 animal for which transmission occurred shed nearly exclusively Alpha. This suggests that within one host, Alpha was marginally more successful at outcompeting Delta in the oropharyngeal cavity.

We then assessed virus sequences in lungs and nasal turbinates to understand if the selective pressure is influenced by spatial dynamics. In Donor lungs, the percentage of Alpha was marginally higher on 5 DPI (median = 42.3% Delta, range = 23.3–75.7%). In the Sentinels groups, either Alpha or Delta outcompeted the other variant within each animal, only one animal (Sentinel 1.8) revealing both variants >15%. In N=5/8 Sentinels 1, yet only in N=1/4 Sentinel 2 animals, Delta outcompeted Alpha. Sequencing of virus isolated from nasal turbinates reproduced this pattern. In Donors, neither variant demonstrated a completive advantage (median = 51.2% Delta, range = 38.7–89.3%). In N=5/8 Sentinels 1, and N=3/8 Sentinels 2, Delta outcompeted Alpha. Combined a trend, while not significant, was observed for increased replication of Delta after the first transmission event, but not after the second, and in the oropharyngeal cavity (swabs) as opposed to lungs (*Figure 5H*) (Donors compared to Sentinels 1: p=0.0559; Donors compared to Sentinels 2: p = >0.9999; Kruskal Wallis test, followed by Dunn's test). Swabs taken at 2 DPI/DPE did significantly predict variant patterns in swabs on 5 DPI/DPE (Spearman's *r*=0.623, p=0.00436) and virus competition in the lower respiratory tract (Spearman's *r*=0.60, p=0.00848). Oral swab samples taken on day 5 strongly correlate with both upper (Spearman's *r*=0.816, p=0.00001) and lower respiratory tract tissue samples (Spearman's *r*=0.832, p=0.00002) taken on the same day (*Figure 5I*).

## Discussion

In immunologically naive humans, peak SARS-CoV-2 shedding occurs multiple days after exposure and in some cases multiple days before onset of symptoms (*Jones et al., 2021*). It is not known how this informs the window of transmissibility, which is poorly understood and difficult to study in the absence of controlled exposures. Measuring the quantity of exhaled virus and size distribution of airborne particles can provide additional insight into the window of transmissibility beyond simply measuring infectious virus in upper respiratory tract swabs. In addition, the shedding of virus in large and fine aerosols may be a function of physiological changes after infection. Past studies in hamsters have shown that SARS-CoV-2 transmissibility is limited to the first 3 days. This coincides with peak shedding and ends before the onset of weight loss, clinical manifestation, and loss of infectious virus shedding in the upper respiratory tract (*Port et al., 2022*; *Sia et al., 2020*; *Rosenke et al., 2020*). We set out to determine if SARS-CoV-2 infection affects host-derived determinants of airborne transmission efficiency early after infection, which may explain this restriction.

Human studies have found similar peak viral RNA levels for Alpha and Delta (*Kissler et al., 2021*; *Elie et al., 2022*) despite their epidemiological differences, including Delta's higher transmissibility (*Earnest et al., 2022*), shorter generation interval (*Hart et al., 2022*), and greater risk of severe disease (*Twohig et al., 2022*). We observe similar kinetics in a controlled experimental setting using the hamster model. We found that swab viral load measurements are a valuable imperfect proxy for the magnitude and timing of airborne shedding. Crucially, there is a period early in infection (around 24 hr post-infection in inoculated hamsters) when oral swabs show high infectious virus titers, but air

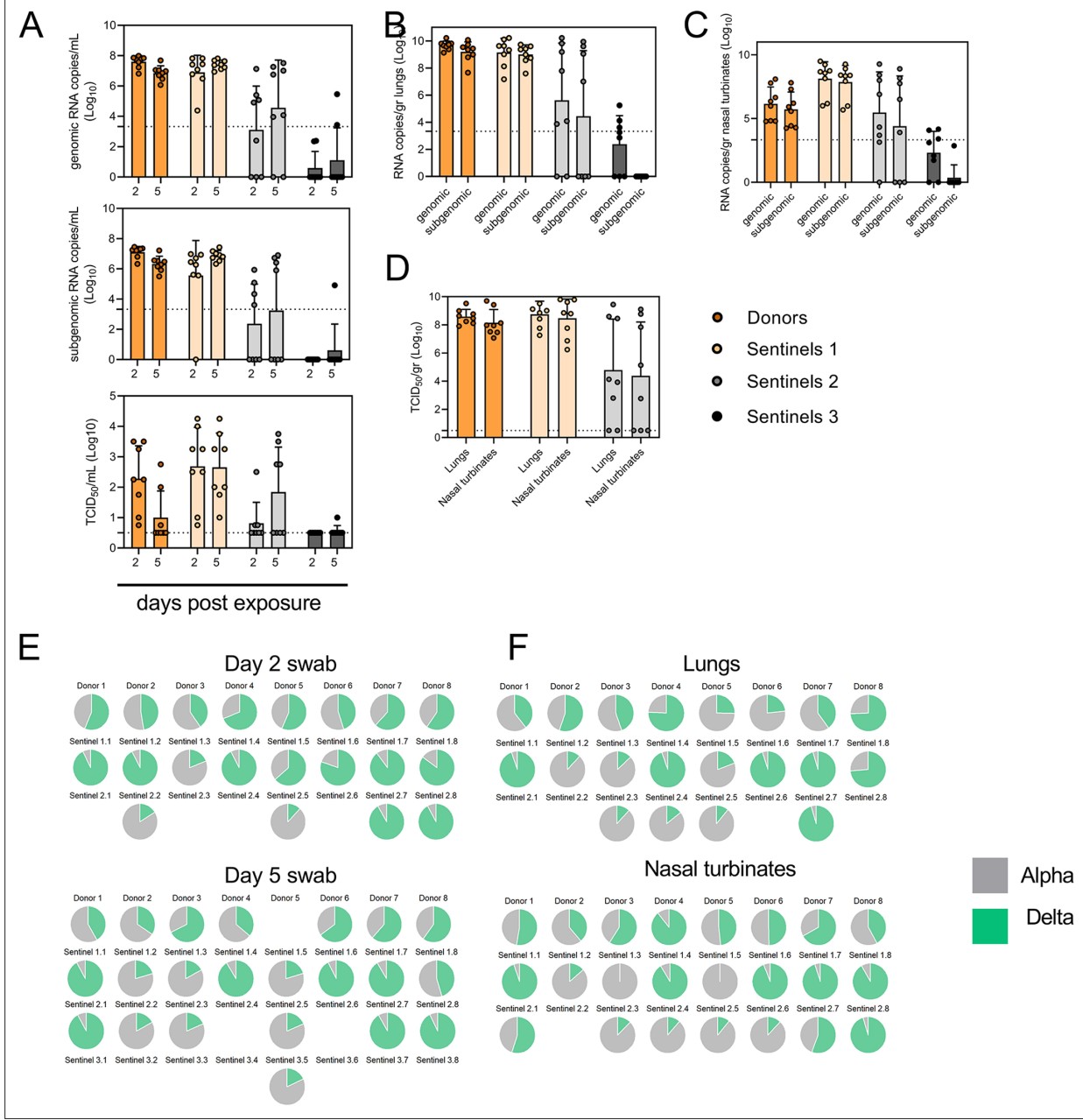

**Figure 5.** Airborne competitiveness of Alpha and Delta SARS-CoV-2 variants. (**A**) Schematic. Donor animals (N=8) were inoculated with Alpha and Delta variant with $5\times10^2$ TCID$_{50}$, respectively, via the intranasal route (1:1 ratio), and three groups of sentinels (Sentinels 1, 2, and 3) were exposed subsequently at a 16.5 cm distance. Animals were exposed at a 1:1 ratio; exposure occurred on day 1 (Donors → Sentinels 1) and day 2 (Sentinels → Sentinels). (**B**) Respiratory shedding measured by viral load in oropharyngeal swabs; measured by gRNA, sgRNA, and infectious titers on days 2 and day 5 post exposure. Bar-chart depicting median, 96% CI and individuals, N=8, ordinary two-way ANOVA followed by Šídák's multiple comparisons test. (**C-E**). Corresponding gRNA, sgRNA, and infectious virus in lungs and nasal turbinates sampled five days post exposure. Bar-chart depicting median, 96% CI and individuals, N=8, ordinary two-way ANOVA, followed by Šídák's multiple comparisons test. Dark orange = Donors, light orange = Sentinels 1, grey = Sentinels 2, dark grey = Sentinels 3, p-values indicated where significant. Dotted line = limit of detection. (**F**) Percentage of Alpha and Delta detected in oropharyngeal swabs taken at days 2 and day 5 post exposure for each individual donor and sentinel, determined by deep sequencing. Pie-charts depict individual animals. Grey = Alpha, teal = Delta. (**G**) Lung and nasal turbinate samples collected on day 5 post inoculation/exposure. (**H**). Summary of data of variant composition, violin plots depicting median and quantiles for each chain link (left) and for each set of samples collected (right). Shading indicates majority of variant (grey = Alpha, teal = Delta). (**I**) Correlation plot depicting spearman r for each chain link (right, day 2 swab) and for each set of samples collected across all animals (left). Colors refer to legend on right. Abbreviations: TCID, Tissue Culture Infectious Dose.

The online version of this article includes the following figure supplement(s) for figure 5:

**Figure supplement 1.** Early virus shedding in donors and sentinels.

samples show low or undetectable levels of virus. Viral shedding should not be treated as a single quantity that rises and falls synchronously throughout the host; spatial models of infection may be required to identify the best correlates of airborne infectiousness (*Snedden et al., 2021*). Attempts to quantify an individual's airborne infectiousness from swab measurements should thus be interpreted with caution, and these spatiotemporal factors should be considered carefully.

While past studies have used whole body plethysmography to differentiate the impact of VOCs on lung function, these have mostly focused on using mathematically derived parameters such as Penh, to compare significant differences on pathology in late acute infection (*Halfmann et al., 2022*). Within our experimental setup we observed high variation within and between different hamsters. Observed differences could be contributed to the behavioral state which correlated with sex, highlighting that future studies of this nature may require increased acclimatization of the animals to these experimental procedures. However, we did observed changes in breathing patterns as early as 2 DPI, preceding clinical symptoms, but coinciding with the window of time when infectious virus was detected in the air.

The majority of SARS-CoV-2 exhaled from hamsters was observed within droplet nuclei <5 µm in size (*Kissler et al., 2021*). We report a rise in <0.53 µm particles and a drop in particles in the 1–10 µm range after infection. One of the caveats of these measurements in small animals is that detected particles may come from aerosolized fomites, and residual dust generated by movement (*Asadi et al., 2021*). In our system, we did not detect any particles originating from dead animals or the environment, but we also saw a noticeable reduction of particles across sizes when movement was minimal, or animals were deeply asleep. Considering the individual variability in the lung function data, we did not observe that this shift in particle production was accompanied by a consistent change in either breathing frequency, tidal volume, or minute volume. It remains to be determined how well airway and particle size distribution dynamics in Syrian hamsters model those in humans. Humans with COVID-19 have been shown to exhale fewer particles than uninfected individuals during normal breathing, but not during coughing (*Viklund et al., 2022*) and fine aerosols have been found to be the major source of virus-loaded droplets. This suggests that a shorter duration of measurable infectious virus in air, as opposed to the upper respiratory tract, could be partially due to early changes in airway constriction and a reduction in exhaled particles of the optimal size range for transmission. The mechanisms involved in the changing aerodynamic particle profile, and the distribution of viral RNA across particle sizes, require further characterization in the hamster model.

Lastly, we compared the transmission efficiency of the Alpha and Delta variants in this system. We did not find a clear transmission advantage for Delta over Alpha in Syrian hamsters, in either an attack rate simulation or when comparing intra- and inter-host competitiveness over multiple generations of airborne transmission. This contrasts sharply with epidemiological observations in the human population, where Delta rapidly replaced Alpha (and other VOCs). The Syrian hamster model may not completely recapitulate all aspects of SARS-CoV-2 virus kinetics and transmission in humans, particularly as the virus continues to adapt to its human host.

Moreover, at the time of emergence of Delta, a large part of the human population was either previously exposed to and/or vaccinated against SARS-CoV-2; that underlying host immune landscape also affects the relative fitness of variants. Our naïve animal model does not capture the high prevalence of pre-existing immunity present in the human population and may therefore be less relevant for studying overall variant fitness in the current epidemiological context. Analyses of the cross-neutralization between Alpha and Delta suggest subtly different antigenic profiles (*van Doremalen et al., 2022*), and Delta's faster kinetics in humans may have also helped it cause more reinfections and 'breakthrough' infections (*Yang and Shaman, 2022*).

Our two transmission experiments yielded different outcomes. When sentinel hamsters were sequentially exposed, first to Alpha and then to Delta, generally no dual infections—both variants detectable—were observed. In contrast, when we exposed hamsters simultaneously to one donor infected with Alpha and another infected with Delta, we were able to detect mixed-variant virus populations in sentinels in one of the cages (Cage F, *Appendix 1—figures 2–4*). The fact that we saw both single-lineage and multi-lineage transmission events suggests that virus population bottlenecks at the point of transmission do indeed depend on exposure mode and duration, as well as donor host shedding. Notably, our analysis suggests that the Alpha-Delta co-infections observed in the Cage F sentinels could be due to that being the one cage in which both the Alpha and the Delta donor

shed substantially over the course of the exposure (Appendix *Appendix 1—figures 2 and 3*). Mixed variant infections were not retained equally, and the relative variant frequencies differed between investigated compartments of the respiratory tract, suggesting roles for randomness or host-and-tissue specific differences in virus fitness.

A combination of host, environmental and virus parameters, many of which vary through time, play a role in virus transmission. These include virus phenotype, shedding in air, individual variability and sex differences, changes in breathing patterns, and droplet size distributions. Alongside recognized social and environmental factors, these host and viral parameters might help explain why the epidemiology of SARS-CoV-2 exhibits classic features of over-dispersed transmission (*Lloyd-Smith et al., 2005*). Namely, SARS-CoV-2 circulates continuously in the human population, but many transmission chains are self-limiting, while rarer superspreading events account for a substantial fraction of the virus's total transmission. Heterogeneity in the respiratory viral loads is high and some infected humans release tens to thousands of SARS-CoV-2 virions/min (*Chen et al., 2021*; *Majra et al., 2021*). Our findings recapitulate this in an animal model and provide further insights into mechanisms underlying successful transmission events. Quantitative assessment of virus and host parameters responsible for the size, duration and infectivity of exhaled aerosols may be critical to advance our understanding of factors governing the efficiency and heterogeneity of transmission for SARS-CoV-2, and potentially other respiratory viruses. In turn, these insights may lay the foundation for interventions targeting individuals and settings with high risk of superspreading, to achieve efficient control of virus transmission (*Kain et al., 2020*).

## Materials and methods

### Cells and viruses

SARS-CoV-2 variant Alpha (B.1.1.7) (hCoV320 19/England/204820464/2020, EPI_ISL_683466) was obtained from Public Health England via BEI Resources. Variant Delta (B.1.617.2/) (hCoV-19/USA/KY-CDC-2-4242084/2021, EPI_ISL_1823618) was obtained from BEI Resources. Virus propagation was performed in VeroE6 cells (kindly provided by Ralph Baric, University of North Carolina, Chapel Hill, USA; also available as VERO C1008 from ATCC (CRL-1586, https://www.atcc.org/products/all/crl-1586.aspx)) in DMEM supplemented with 2% fetal bovine serum, 1 mM L-glutamine, 50 U/mL penicillin and 50 µg/mL streptomycin (DMEM2). VeroE6 cells were maintained in DMEM supplemented with 10% fetal bovine serum, 1 mM L- glutamine, 50 U/mL penicillin and 50 µg/ml streptomycin. No mycoplasma and no contaminants were detected. All virus stocks were sequenced; no SNPs compared to the patient sample sequence were detected in the Delta stock. In the Alpha stock we detected: ORF1AB D3725G: 13% ORF1AB L3826F: 18%.

### Pseudotype entry assay

The spike coding sequences for SARS-CoV-2 variant Alpha and Delta were truncated by deleting 19 aa at the C-terminus. The S proteins with the 19 aa deletions of coronaviruses were previously reported to show increased efficiency incorporating into virions of VSV (*Fukushi et al., 2005*; *Kawase et al., 2009*). These sequences were codon optimized for human cells, then appended with a 5′ kozak expression sequence (GCCACC) and 3′ tetra-glycine linker followed by nucleotides encoding a FLAG-tag sequence (DYKDDDDK). These spike sequences were synthesized and cloned into pcDNA3.1⁺(GenScript). Human and hamster ACE2 (Q9BYF1.2 and GQ262794.1, respectively) were synthesized and cloned into pcDNA3.1⁺ (GenScript). All DNA constructs were verified by Sanger sequencing (ACGT). BHK cells were seeded in black 96-well plates and transfected the next day with 100 ng plasmid DNA encoding human or hamster ACE2, using polyethylenimine (Polysciences). All downstream experiments were performed 24 hr post-transfection. Pseudotype production was carried out as described previously (*Letko et al., 2020*). Briefly, plates pre-coated with poly-L-lysine (Sigma–Aldrich) were seeded with 293T cells and transfected the following day with 1200 ng of empty plasmid and 400 ng of plasmid encoding coronavirus spike or no-spike plasmid control (green fluorescent protein (GFP)). After 24 hr, transfected cells were infected with VSVΔG seed particles pseudotyped with VSV-G as previously described (*Letko et al., 2020*; *Takada et al., 1997*). After 1 hr of incubating with intermittent shaking at 37 °C, cells were washed four times and incubated in 2 mL DMEM supplemented with 2% FBS, penicillin/streptomycin, and L-glutamine for 48 hr. Supernatants

were collected, centrifuged at 500 x *g* for 5 min, aliquoted, and stored at −80 °C. BHK cells previously transfected with ACE2 plasmids of interest were inoculated with equivalent volumes of pseudotype stocks. Plates were then centrifuged at 1200 x *g* at 4 °C for 1 hr and incubated overnight at 37 °C. Approximately 18–20 hr post-infection, Bright-Glo luciferase reagent (Promega) was added to each well, 1:1, and luciferase was measured. Relative entry was calculated by normalizing the relative light unit for spike pseudotypes to the plate relative light unit average for the no-spike control. Each figure shows the data for two technical replicates.

## Structural interaction analysis

The locations of the described spike mutations in the Alpha and Delta VOCs were highlighted on the SARS-CoV-2 spike structure (PDB 6ZGE, *Wrobel et al., 2020*). To visualize the molecular interactions at the RBD-ACE2 binding interface, the crystal structure of the Alpha variant RBD and human ACE2 complex (PDB 7EKF *Han et al., 2021*) was utilized. All figures were generated using The PyMOL Molecular Graphics System (https://www.schrodinger.com/pymol).

## Aerosol caging

Aerosol cages as described by *Port et al., 2022* were used for transmission experiments and air sampling as indicated. The aerosol transmission system consisted of plastic hamster boxes (Lab Products) connected by a plastic tube. The boxes were modified to accept a 7.62 cm (3') plastic sanitary fitting (McMaster-Carr), which enabled the length between the boxes to be changed. Airflow was generated with a vacuum pump (Vacuubrand) attached to the box housing the naïve animals and was controlled with a float-type meter/valve (McMaster-Carr).

## Experimental study design for in vivo studies

Sample size was determined based on expected differences in virological parameters (0.5 log difference in shedding or tissue titers) or transmission efficiency (75% difference or 2.25 ratio). Animals were randomly assigned to groups, keeping them sex separated. Experimenters performing data analysis were blinded where possible. No animals were excluded from the analysis.

## Hamster infection with Alpha and Delta

Four-to-6-week-old female and male Syrian hamsters (ENVIGO) were inoculated (10 animals per virus) intranasally (IN) with either SARS-CoV-2 variant Alpha (B.1.1.7) (hCoV320 19/England/204820464/2020, EPI_ISL_683466), variant Delta (B.1.617.2/) (hCoV-19/USA/KY-CDC-2-4242084/2021, EPI_ISL_1823618)., or no virus (anesthesia controls). IN inoculation was performed with 40 μL sterile DMEM containing $10^3$ TCID$_{50}$ SARS-CoV-2 or simply sterile DMEM. At 5 days post inoculation (DPI), five hamsters from each group were euthanized and tissues were collected. The remaining five animals were euthanized at 14 DPI for disease course assessment and shedding analysis. For the control group, no day 5 necropsy was performed. Hamsters were weighed daily, and oropharyngeal swabs were taken on days 1, 2, 3, 4, 5, and 7. Swabs were collected in 1 mL DMEM with 200 U/mL penicillin and 200 μg/mL streptomycin. For the control group, mock swabs were performed to ensure animals underwent the same anesthesia protocols as infection groups. On days –1, 0, 1, 2, 3, 4, 5, 6, 7, and 14 whole body plethysmography was performed. Profiles of particles produced by hamsters were collected on days 0, 1, 3, and 5. Cage air was sampled on day 0, 1, 2, 3, 4, and 5. Hamsters were observed daily for clinical signs of disease. Necropsies and tissue sampling were performed according to IBC-approved protocols.

## Air sampling of hamster cages

During the first 5 days, hamsters were housed in modified aerosol cages (only one hamster box) hooked up to an air pump. Air flow was generated at 30 cage changes/hr. Between the cage and the pump a 47 mm gelatin air filter was installed. Filters were exchanged in 24 hr intervals. The filters were dissolved in 5 mL of DMEM containing 10% FBS and presence of virus was determined by qRT PCR and plaque assay.

## Aerodynamic particle sizing of exhaled droplets

Two strategies were used to measure the aerodynamic diameter of droplets exhaled by hamsters. SARS-CoV-2 inoculated hamsters or uninfected control animals were placed into a 1.25 L isoflurane

chamber. This allowed free movement of the animal in the chamber. The chamber was hooked up with one port to a HEPA filter. The second port was hooked up to a Model 3321 aerodynamic particle sizer spectrometer (TSI). Both chamber and particle sizer were placed into a BSC class II cabinet. Animals remained in the chamber for 5x1 min readings. For each set of readings, there were 52 different particle sizes. For each hamster and timepoint, the total number of particles was calculated and the percent of particles in a particular diameter range was derived using this total. RStudio 2021.09.1 Build 372 Ghost Orchid Release, R version 4.1.2 (2021-11-01), Tidyverse R package version 1.3.1 (2021-04-15), and Emmeans R package version 1.7.2 (2022-01-04) were used for the aerodynamic particle size analysis.

To differentiate between particle profiles produced by an awake and moving animal and those produced by a sleeping animal with limited movement, uninfected age-matched hamsters (three males and two females) were acclimatized to being inside a 38.1 mm inside diameter tube hooked up to a particle sizer (*Figure 5A and B*). Both tube and particle sizer were placed into a BSC class II cabinet. To acclimate the animals to the tube, sunflower seeds were provided to encourage investigation and free entry and exit from the tube. After animals became used to being in the tube, ends were capped as depicted and 5x5 min readings were taken. The particle size was measured using a Model 3321 aerodynamic particle sizer spectrometer (TSI). Particle size profiles were analyzed using TSI software. As a control, particles originating from empty enclosures and euthanized animals were recorded and found to be absent.

## Aerosol transmission attack rate experiment

Four-to-6-week-old female and male Syrian hamsters (ENVIGO) were used. In this experiment naïve hamsters (sentinels) were exposed to donors infected with either Alpha or Delta in the same aerosol transmission set-up to evaluate the attack rates of both variants. Donor hamsters were infected intranasally as described above with $10^3$ TCID$_{50}$ SARS-CoV-2 (Alpha or Delta, N=7, respectively) and housed individually. After 24 hr, donor animals were placed into the donor cage. 4 or 5 sentinels were placed into the sentinel cage (N=34, 7 iterations), which was connected to the donor cage by a 2 m tube and exposed for 4 hr. Air flow was generated between the cages from the donor to the sentinel cage at 30 cage changes/h. One donor inoculated with Alpha, and one donor inoculated with Delta were randomly chosen for each scenario. Both donors were either placed together into the donor cage, or, alternatively, first one donor was placed into the cage for 2 hr, then the other for 2 hr. To ensure no cross-contamination, the donor cages and the sentinel cages were never opened at the same time, sentinel hamsters were not exposed to the same handling equipment as donors, and equipment was disinfected with either 70% ETOH or 5% Microchem after each sentinel. Regular bedding was replaced by alpha-dri bedding to avoid the generation of dust particles. Oropharyngeal swabs were taken for donors after completion of the exposure and for sentinels on days 2, 3, and 5 after exposure. Swabs were collected in 1 mL DMEM with 200 U/mL penicillin and 200 µg/mL streptomycin. Donors were euthanized after exposure ended, and sentinels were euthanized on day 5 for collection of lungs. All animals were always single housed outside the exposure window.

## Variant competitiveness transmission chain

Four-to six-week-old female and male Syrian hamsters (ENVIGO) were used. Donor hamsters (N=8) were infected intranasally as described above with $10^3$ TCID$_{50}$ SARS-CoV-2 at a 1:1 ratio of Alpha and Delta (exact titration of the inoculum for both variants = 503 TCID$_{50}$, 80% Delta sequencing reads). After 12 hr, donor animals were placed into the donor cage and sentinels (Sentinels 1, N=8) were placed into the sentinel cage (1:1) at a 16.5 cm distance with an airflow of 30 cage changes/h as described by *Port et al., 2022*. Hamsters were co-housed for 24 hr. The following day, donor animals were re-housed into regular rodent caging. One day later, Sentinels 1 were placed into the donor cage of new transmission set-ups. New sentinels (Sentinels 2, N=8) were placed into the sentinel cage at a 16.5 cm distance with an airflow of 30 changes/hr. Hamsters were co-housed for 24 hr. Then, Sentinels 1 were re-housed into regular rodent caging and Sentinels 2 were placed into the donor cage of new transmission set-ups 1 day later. New sentinels (Sentinels 3, N=8) were placed into the sentinel cage at a 16.5 cm distance with an airflow of 30 changes/hr. Hamsters were co-housed for 24 hr. Then both Sentinels 2 and Sentinels 3 were re-housed to regular rodent caging and monitored until 5 DPE. Oropharyngeal swabs were taken for all animals at 2 and 5 DPI/DPE. All animals were euthanized at

5 DPI/DPE for collection of lung tissue and nasal turbinates. To ensure no cross-contamination, the donor cages and the sentinel cages were never opened at the same time, sentinel hamsters were not exposed to the same handling equipment as donors, and the equipment was disinfected with either 70% EtOH or 5% Microchem after each sentinel. Regular bedding was replaced by alpha-dri bedding to avoid the generation of dust particles.

## Within-host kinetics model

We used Bayesian inference to fit a semi-mechanistic model of within-host virus kinetics and shedding to our data from inoculated hamsters. Briefly, the model assumes a period of exponential growth of virus within the host up to a peak viral load, followed by exponential decay. It assumes virus shedding into the air follows similar dynamics, and the time of peak air shedding and peak swab viral load may be offset from each other by an inferred factor. Decay of RNA may be slower than that of infectious virus by an inferred factor, representing the possibility, seen in our data, that some amplified RNA may be residual rather than representative of current infectious virus levels. We also inferred conversion factors (ratios) among the various quantities, that is how many oral swab sgRNA copies correspond to an infectious virion at peak viral load. We fit the model to our swab and cage air sample data using Numpyro (*Phan et al., 2019*), which implements a No-U-Turn Sampler (*Hoffman and Gelman, 2014*). For full mathematical details of the model and how it was fit, including prior distribution choices and predictive checks (*Appendix 1—figure 1*), see Appendix: Within-host dynamics model and Bayesian inference methods.

## Whole body plethysmography

Whole body plethysmography was performed on SARS-CoV-2 and uninfected Syrian hamsters. Animals were individually acclimated to the plethysmography chamber (Buxco Electronics Ltd., NY, USA) for 20 min, followed by a 5-min measurement period with measurements taken continuously and averaged over two-second intervals. Initial data was found to have an especially high rejection index (Rinx) for breaths, so was reanalyzed using a custom Buxco formula to account for differences between mice and hamsters. This included expanding the acceptable balance range, the percent change in volume between inhalation and exhalation, from 20–180% to 15–360%. Reanalysis using this algorithm resulted in the Rinx across all hamsters from one day before infection to 5 days post-infection decreasing from 62.97% to 48.65%. The reanalyzed data were then used for further analysis. Each hamster's individual averages one day prior to infection were used as their baselines for data analysis.

Areas under the curve (AUCs) for each parameter were calculated for each individual hamster based on their raw deviation from baseline at each time point. Either positive or negative peaks were assessed based on parameter-specific changes. Principal component analyses (PCAs) to visualize any potential clustering of animals over the course of infection were performed for each day on raw values for each of the parameters to accurately capture the true clustering with the least amount of data manipulation. PCAs and associated visualizations were coded in R using RStudio version 1.4.1717 (*RStudio Team, 2021*). The readxl package version 1.3.1 was then used to import Excel data into RStudio for analysis (*Wickham and Bryan, 2023*). Only parameters that encapsulated measures of respiratory function were included (zero-centered, scaled). The factoextra package version 1.0.1 (*Kassambara and Mundt, 2020*) was used to determine the optimal number of clusters for each PCA via the average silhouette width method and results were visualized using the ggplot2 package (*Wickham, 2016*). Correlation plots were generated based on raw values for each lung function parameter using the corrplot package version 0.90 (*Wei and Simko, 2021*). The color palette for correlation plots was determined using RColorBrewer version 1.1–2 (*Neuwirth, 2022*).

## Viral RNA detection

Swabs from hamsters were collected as described above. A total of 140 µL was utilized for RNA extraction using the QIAamp Viral RNA Kit (Qiagen) using QIAcube HT automated system (Qiagen) according to the manufacturer's instructions with an elution volume of 150 µL. For tissues, RNA was isolated using the RNeasy Mini kit (Qiagen) according to the manufacturer's instructions and eluted in 60 µL. Sub-genomic (sg) and genomic (g) viral RNA were detected by qRT-PCR (*Corman et al., 2020*). RNA was tested with TaqMan Fast Virus One-Step Master Mix (Applied Biosystems) using

QuantStudio 6 or 3 Flex Real-Time PCR System (Applied Biosystems). SARS-CoV-2 standards with known copy numbers were used to construct a standard curve and calculate copy numbers/mL or copy numbers/g. Limit of detection = 10 copies/rxn.

## Viral titration

Viable virus in tissue samples was determined as previously described *Yang et al., 2021*. In brief, lung tissue samples were weighed, then homogenized in 1 mL of DMEM (2% FBS). Swabs were used undiluted. VeroE6 cells were inoculated with ten-fold serial dilutions of homogenate, incubated for 1 hr at 37 °C, and the first two dilutions washed twice with 2% DMEM. For swab samples, cells were inoculated with ten-fold serial dilutions and no wash was performed. After 6 days, cells were scored for cytopathic effect. $TCID_{50}$/mL was calculated by the Spearman-Karber method. To determine titers in air samples, a plaque assay was used. VeroE6 cells were inoculated with 200 µL/well (48-well plate) of undiluted samples, no wash was performed. Plates were spun for 1 hr at RT at 1000 rpm. A total of 800 µL of CMC (500 mL MEM Cat#10370, Gibco, must contain NEAA), 5 mL PenStrep, 7.5 g carboxymethylcellulose (CMC, Cat# C4888, Sigma, sterilize in autoclave) overlay medium was added to each well and plates incubated for 6 days at 37 °C. Plates were fixed with 10% formalin overnight, then rinsed and stained with 1% crystal violet for 10 min. Plaques were counted.

## Serology

Serum samples were analyzed as previously described (*Yinda et al., 2021*). In brief, maxisorp plates (Nunc) were coated with 50 ng spike protein (generated in-house, purified recombinant) per well. Plates were incubated overnight at 4 °C. Plates were blocked with casein in phosphate buffered saline (PBS) (Thermo Fisher) for 1 hr at room temperature (RT). Serum was diluted twofold in blocking buffer and samples (duplicate) were incubated for 1 hr at RT. Secondary goat anti-hamster IgG Fc (horseradish peroxidase (HRP)-conjugated, Cat.No. 5220–0371 Lot. 10492253, Seracare) antibodies were used for detection and KPL TMB 2-component peroxidase substrate kit (SeraCare, Cat.No. 5120–0047) was used for visualization. The reaction was stopped with KPL stop solution (Seracare) and plates were read at 450 nm. The threshold for positivity was calculated as the average plus 3 x the standard deviation of negative control hamster sera.

## MesoPlex assay

The V-PLEX SARS-CoV-2 Panel 13 (IgG) kit from Meso Scale Discovery was used to test binding antibodies against spike protein of SARS-CoV-2 with 10,000-fold diluted serum obtained from hamsters 14 DPI. A standard curve of pooled hamster sera positive for SARS-CoV-2 spike protein was serially diluted fourfold. The secondary antibody was prepared by conjugating a goat anti-hamster IgG cross-adsorbed secondary antibody (Thermo Fisher, Cat.No. SA5-10284) using the MSD GOLD SULFO-TAG NHS-Ester Conjugation Pack (MSD). The secondary antibody was diluted 10,000 X for use on the assay. The plates were prepped, and samples were run according to the kit's instruction manual. After plates were read by the MSD instrument, data was analyzed with the MSD Discovery Workbench Application.

## Virus neutralization

Heat-inactivated γ-irradiated sera were two-fold serially diluted in DMEM. 100 $TCID_{50}$ of SARS-CoV-2 variant Alpha (B.1.1.7) (hCoV320 19/England/204820464/2020, EPI_ISL_683466) or variant Delta (B.1.617.2/) (hCoV-19/USA/KY-CDC-2-4242084/2021, EPI_ISL_1823618) was added. After 1 hr of incubation at 37 °C and 5% $CO_2$, the virus:serum mixture was added to VeroE6 cells. CPE was scored after 5 days at 37 °C and 5% $CO_2$. The virus neutralization titer was expressed as the reciprocal value of the highest dilution of the serum that still inhibited virus replication. The antigenic map was constructed as previously described (*Smith et al., 2004*; *Fonville et al., 2014*) using the antigenic cartography software from https://acmacs-web.antigenic-cartography.org. In brief, this approach to antigenic mapping uses multidimensional scaling to position antigens (viruses) and sera in a map to represent their antigenic relationships. The maps here relied on the first SARS-CoV-2 infection serology data of Syrian hamsters. The positions of antigens and sera were optimized in the map to minimize the error between the target distances set by the observed pairwise virus-serum combinations. Maps were

effectively constructed in only one dimension because sera were only titrated against two viruses and the dimensionality of the map is constrained to the number of test antigens minus one.

## Next-generation sequencing of virus

Total RNA was extracted from oral swabs, lungs, and nasal turbinates using the Qia Amp Viral kit (Qiagen, Germantown, MD), eluted in EB, and viral Ct values were calculated using real-time PCR. Subsequently, 11 µL of extracted RNA was used as a template in the ARTIC nCoV-2019 sequencing protocol V.1 (Protocols.io - https://www.protocols.io/view/ncov-2019-sequencing-protocol-bbmuik6w) to generate 1st-strand cDNA. Five microliters were used as template for Q5 HotStart Pol PCR (Thermo Fisher Sci, Waltham, MA) together with 10 µM stock of a single primer pair from the ARTIC nCoV-2019 v3 Panel (Integrated DNA Technologies, Belgium), specifically 76 L_alt3 and 76 R_alt0. Following 35 cycles and 55 °C annealing temperature, products were AmPure XP cleaned and quantitated with Qubit (Thermo Fisher Scientific) fluorometric quantitation per instructions. Following visual assessment of 1 µL on a Tape Station D1000 (Agilent Technologies, Santa Clara, CA), a total of 400 ng of product was taken directly into TruSeq DNA PCR-Free Library Preparation Guide, Revision D (Illumina, San Diego, CA) beginning with the Repair Ends step (q.s. to 60 µL with RSB). Subsequent cleanup consisted of a single 1:1 AmPure XP/reaction ratio and all steps followed the manufacturer's instructions including the Illumina TruSeq CD (96) Indexes. Final libraries were visualized on a BioAnalyzer HS chip (Agilent Technologies) and quantified using KAPA Library Quant Kit - Illumina Universal qPCR Mix (Kapa Biosystems, Wilmington, MA) on a CFX96 Real-Time System (BioRad, Hercules, CA). Libraries were diluted to 2 nM stock, pooled together in equimolar concentrations, and sequenced on the Illumina MiSeq instrument (Illumina) as paired-end 2X250 base pair reads. Because of the limited diversity of a single-amplicon library, 20% PhiX was added to the final sequencing pool to aid in final sequence quality. Raw fastq reads were trimmed of Illumina adapter sequences using cutadapt version 1.1227, then trimmed and filtered for quality using the FASTX-Toolkit (Hannon Lab, CSHL). To process the ARTIC data, a custom pipeline was developed (*Avanzato et al., 2020*). Fastq read pairs were first compared to a database of ARTIC primer pairs to identify read pairs that had correct, matching primers on each end. Once identified, the ARTIC primer sequence was trimmed off. Read pairs that did not have the correct ARTIC primer pairs were discarded. Remaining read pairs were collapsed into one sequence using AdapterRemoval (*Schubert et al., 2016*) requiring a minimum 25 base overlap and 300 base minimum length, generating ARTIC amplicon sequences. Identical amplicon sequences were removed, and the unique amplicon sequences were then mapped to the SARS-CoV-2 genome (MN985325.1) using Bowtie2 (*Langmead and Salzberg, 2012*). Aligned SAM files were converted to BAM format, then sorted and indexed using SAMtools (*Li et al., 2009*). Variant calling was performed using Genome Analysis Toolkit (GATK, version 4.1.2) HaplotypeCaller with ploidy set to 2 (*McKenna et al., 2010*). Single nucleotide polymorphic variants were filtered for QUAL >200 and quality by depth (QD) >20 and indels were filtered for QUAL >500 and QD >20 using the filter tool in bcftools, v1.9 (*Li et al., 2009*). Pie charts were generated using ggplot2 (*Wickham, 2016*) in R 4.2.1 using RStudio version 1.4.1717 (*RStudio Team, 2021*).

## Histopathology

Necropsies and tissue sampling were performed according to IBC-approved protocols. Tissues were fixed for a minimum of 7 days in 10% neutral buffered formalin with 2 changes. Tissues were placed in cassettes and processed with a Sakura VIP-6 Tissue Tek on a 12 hr automated schedule using a graded series of ethanol, xylene, and ParaPlast Extra. Prior to staining, embedded tissues were sectioned at 5 µm and dried overnight at 42 °C. Using GenScript U864YFA140-4/CB2093 NP-1 (1:1000), specific anti-CoV immunoreactivity was detected using the Vector Laboratories ImPress VR anti-rabbit IgG polymer (# MP-6401) as secondary antibody. The tissues were then processed using the Discovery Ultra automated processor (Ventana Medical Systems) with a ChromoMap DAB kit Roche Tissue Diagnostics (#760–159).

## Statistical analysis

Significance tests were performed as indicated where appropriate for the data using GraphPad Prism 9. Unless stated otherwise, statistical significance levels were determined as follows: ns = $p > 0.05$; *=$p \leq 0.05$; **=$p \leq 0.01$; ***=$p \leq 0.001$; ****=$p \leq 0.0001$. Exact nature of tests is stated where appropriate.

Data collected from animal experiments was assumed non-parametric and tests were applied appropriately. All data collected from animal experiments represents biological replicates.

## Acknowledgements

We would like to thank Ryan Stehlik, Seth Cooley and Shanda Sarchette, and the animal care takers for their assistance during the study. The following reagent was obtained through BEI Resources, NIAID, NIH: SARS-CoV-2 variant Alpha (B.1.1.7) (hCoV320 19/England/204820464/2020, EPI_ISL_683466) and variant Delta (B.1.617.2/) (hCoV-19/USA/KY-CDC-2-4242084/2021, EPI_ISL_1823618). We thank Neeltje van Doremalen, Emmie de Wit, Brandi Williamson, Sujatha Rashid, Ranjan Mukul, Kimberly Stemple, Bin Zhou, Natalie Thornburg, Sue Tong, Stacey Ricklefs, Sarah Anzick for gracefully sharing viruses or propagating and sequencing stocks. We would like to thank Amy Tillman for assistance with the aerodynamic particle data analysis.

## Additional information

### Funding

| Funder | Grant reference number | Author |
| --- | --- | --- |
| National Institutes of Health | 1ZIAAI001179-01 | Vincent J Munster |
| Defense Advanced Research Projects Agency | D18AC00031 | James O Lloyd-Smith |
| National Science Foundation | DEB-1557022 | James O Lloyd-Smith |
| National Institute of Allergy and Infectious Diseases | NIAID SAVE | Vincent J Munster |

The funders had no role in study design, data collection and interpretation, or the decision to submit the work for publication.

### Author contributions

Julia R Port, Conceptualization, Resources, Data curation, Formal analysis, Validation, Investigation, Visualization, Methodology, Writing – original draft, Project administration, Writing – review and editing; Dylan H Morris, Conceptualization, Data curation, Software, Formal analysis, Validation, Investigation, Methodology, Writing – original draft, Writing – review and editing; Jade C Riopelle, Claude Kwe Yinda, Formal analysis, Investigation, Methodology, Writing – review and editing; Victoria A Avanzato, Myndi G Holbrook, Investigation, Methodology, Writing – review and editing; Trenton Bushmaker, Kent Barbian, Validation, Investigation, Visualization, Methodology, Writing – review and editing; Jonathan E Schulz, Taylor A Saturday, Carl Shaia, Validation, Investigation, Methodology, Writing – review and editing; Colin A Russell, Software, Supervision, Validation, Investigation, Methodology, Writing – review and editing; Rose Perry-Gottschalk, Validation, Visualization, Methodology, Writing – review and editing; Craig Martens, Supervision, Validation, Methodology, Writing – review and editing; James O Lloyd-Smith, Resources, Software, Formal analysis, Supervision, Validation, Project administration, Writing – review and editing; Robert J Fischer, Resources, Formal analysis, Supervision, Validation, Investigation, Methodology, Writing – review and editing; Vincent J Munster, Conceptualization, Resources, Data curation, Formal analysis, Supervision, Funding acquisition, Validation, Investigation, Visualization, Writing – original draft, Project administration, Writing – review and editing

### Author ORCIDs

Julia R Port ⓘ https://orcid.org/0000-0002-0489-6591
Claude Kwe Yinda ⓘ http://orcid.org/0000-0002-5195-5478
Trenton Bushmaker ⓘ http://orcid.org/0000-0002-2161-4808
James O Lloyd-Smith ⓘ http://orcid.org/0000-0001-7941-502X
Robert J Fischer ⓘ http://orcid.org/0000-0002-1816-472X

Vincent J Munster 🔟 https://orcid.org/0000-0002-2288-3196

### Ethics

All animal experiments were conducted in an AAALAC International-accredited facility and were approved by the Rocky Mountain Laboratories Institutional Care and Use Committee following the guidelines put forth in the Guide for the Care and Use of Laboratory Animals 8th edition, the Animal Welfare Act, United States Department of Agriculture and the United States Public Health Service Policy on the Humane Care and Use of Laboratory Animals. Protocol number 2021-034-E. Work with infectious SARS-CoV-2 virus strains under BSL3 conditions was approved by the Institutional Biosafety Committee (IBC). For the removal of specimens from high containment areas virus inactivation of all samples was performed according to IBC-approved standard operating procedures.

Reviewer #1 (Public Review): https://doi.org/10.7554/eLife.87094.3.sa1
Reviewer #2 (Public Review): https://doi.org/10.7554/eLife.87094.3.sa2
Author Response https://doi.org/10.7554/eLife.87094.3.sa3

## Additional files

### Supplementary files

• Supplementary file 1. Airborne attack rate of Alpha and Delta SARS-CoV-2 variants. Donor animals (N=7) were inoculated with either the Alpha or Delta variant and paired together randomly in 7 attack rate scenarios (A-G). One day after inoculation, 4–5 sentinels were exposed for a duration of 4 h in an aerosol transmission set-up. Percentage of Alpha and Delta detected in oropharyngeal swabs taken at day 2 and day 5 post exposure by deep sequencing.

• MDAR checklist

### Data availability

Data deposited in Figshare (https://doi.org/10.6084/m9.figshare.20493045) and Github (https://github.com/dylanhmorris/host-viral-determinants, copy archived at *Morris, 2024*).

The following dataset was generated:

| Author(s) | Year | Dataset title | Dataset URL | Database and Identifier |
|---|---|---|---|---|
| Port J, Morris D, Kwe CY, Riopelle JC, Saturday TA, Schulz J, Fischer R, Munster V, Martens C, Perry-Gottschalk R, Barbian K, Holbrook MG, Bushmaker T, Avanzato V, Shaia C, Russel C, Lloyd-Smith JO | 2023 | Host and viral determinants of airborne transmission of SARS-CoV-2 in the Syrian hamster | https://doi.org/10.6084/m9.figshare.20493045 | figshare, 10.6084/m9.figshare.20493045 |

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

## Appendix 1

### Within-host dynamics model and Bayesian inference methods

## Introduction

We modeled the observation process explicitly to extract maximal information from the data generated.

### Notation

In the text that follows, we use the following mathematical notation.

### Logarithms and exponentials

$\log(x)$ denotes the logarithm base $e$ of $x$ (sometimes called $\ln(x)$). We explicitly refer to the logarithm base 10 of $x$ as $\log_{10}(x)$. $\exp(x)$ denotes $e^x$.

### Probability distributions

The symbol ~ denotes that a random variable is distributed according to a given probability distribution. So, for example

$$X \sim \text{Normal}(0, 1)$$

indicates that the random variable $X$ is normally distributed with mean 0 and standard deviation 1.

We parameterize normal distributions as:

$$\text{Normal}(\text{mean}, \text{standard deviation})$$

We parameterize positive-constrained normal distributions (i.e. with lower limit 0) as:

$$\text{PosNormal}(\text{mode}, \text{standard deviation})$$

We parameterize censored normal distributions, in which values outside the censoring range are reported at the lower limit of detection (lld) and upper limit of detection (uld), respectively, as:

$$\text{CensoredNormal}(\text{mean}, \text{standard deviation}, \text{lld}, \text{uld})$$

We parameterize Poisson distributions as:

$$\text{Poisson}(\text{mean})$$

### Units

Unless otherwise stated, we express time in units of hours, volume in units of mL, infectious virus in units of plaque forming units collected on our air filter, and sgRNA in units of copy numbers.

### Dynamics model

We modeled the within-host dynamics of the virus within inoculated hamsters as a process of exponential growth of virus up to a peak, followed by exponential decay of virus down from that peak. The principal quantity of interest is airborne virus shedding over time $V_a(t)$, expressed in units of infectious virions exhaled per unit volume of exhaled air per unit time. We express time in units of hours post-infection.

### Growth and decay of air shedding

We denote the exponential growth rate of the virus within the hamster by $g$ and the exponential decay rate after the peak by $d_{av}$. We denote the time of peak airborne shedding $t_a > 0$ and define $t = 0$ as the time of inoculation.

Our model is therefore:

$$
V_a(t) = \begin{cases} V_a(0) \exp[gt] & t < t_a \\ V_a(0) \exp[gt_a - d_{av}(t - t_a)] & t \geq t_a \end{cases}
$$

(1)

## Offset growth and decay of oral shedding

Since we also took measurements of virus shed in oral swabs, we incorporated the dynamics of oral swab shedding $V_o(t)$ into our model. We modeled $V_o(t)$ as offset in time from the dynamics of airborne shedding $V_a(t)$ by some offset factor $\omega > 0$. That is, the time of peak oral shedding $t_o$ is:

$$t_o = \omega t_a \tag{2}$$

Note that $\omega < 1$ implies that swab shedding peaks earlier than airborne shedding, $\omega > 1$ implies that swab shedding peaks later, and $\omega = 1$ implies the peaks coincide in time.

We also allowed for the possibility that oral virus shedding decays at a faster or slower rate $d_{ov}$ than airborne virus shedding, which decays at a rate $d_{av}$. Specifically, we defined the ratio of $d_{ov}$ to $d_{av}$ as $q_o > 0$, so:

$$d_{ov} = q_o d_{av} \tag{3}$$

Then:

$$V_o(t) = \begin{cases} V_o(0) \exp\left[gt\right] & t < t_o \\ V_o(0) \exp\left[gt_o - d_{ov}(t - t_o)\right] & t \geq t_o \end{cases} \tag{4}$$

## Relationship between sgRNA and infectious virus

We also modeled the possibility that measured sgRNA decays slower or faster than measured infectious virus. Slower decay, for instance, could result from persistence of undegraded RNA after all infectious virions have been neutralized or otherwise lost infectivity.

We modeled this possibly different RNA shedding decay rate with an estimated ratio $q_n > 0$ that relates sgRNA shedding decay to infectious virus shedding decay. So the decay rate of airborne sgRNA shedding $d_{an}$ is:

$$d_{an} = q_n d_{av} \tag{5}$$

And similarly the decay rate of oral sgRNA shedding $d_{on}$ is:

$$d_{on} = q_n d_{ov} = q_o q_n d_{av} \tag{6}$$

We modeled the ratio between produced virus $V(t)$ and produced sgRNA copies $N(t)$ with multipliers $o_n$ for oral swabs and $a_n$ for air samples. So before the decay phase begins, $V(t)$ and $N(t)$ are linearly related.

$$\begin{aligned} N_a(t) && a_n V_a(t) && if\ t < t_a \\ N_o(t) && o_n V_o(t) && if\ t < t_o \end{aligned} \tag{7}$$

The dynamics of airborne sgRNA shedding $N_a(t)$ and oral sgRNA shedding $N_o(t)$ are therefore equivalent to those for infectious virus in *Equation 1* and *Equation 4*, respectively, but with $d_{an}$ instead of $d_{av}$, $d_{on}$ instead of $d_{ov}$, and initial values $N_a(0) = a_n V_a(0)$, $N_o(0) = o_n V_o(0)$:

$$N_a(t) = \begin{cases} N_a(0) \exp\left[gt\right] & t < t_a \\ N_a(0) \exp\left[gt_a - d_{an}(t - t_a)\right] & t \geq t_a \end{cases} \tag{8}$$

$$N_o(t) = \begin{cases} N_o(0) \exp\left[gt\right] & t < t_o \\ N_o(0) \exp\left[gt_o - d_{on}(t - t_o)\right] & t \geq t_o \end{cases} \tag{9}$$

### Initial shedding value

For inference purposes, rather than set a prior distribution on the initial airborne shedding viral load $V_0 = V_a(0)$, we instead set a prior on the peak airborne viral load $V_{max} = V_a(t_a)$ and back-calculated $V_a(0)$ (and thus $V_o(0)$, $N_a(0)$, and $N_o(0)$) via:

$$\log(V_0) = \log(V_{max}) - gt_a \tag{10}$$

### Variant effects

We allowed the two variants of interest, Alpha and Delta, to take on different typical values for all of the virological parameters: $g$, $d_{av}$, $t_a$, $\omega$, $q_o$, $q_n$, $a_n$, $o_n$, and mean peak viral shedding $V_{max}$. We denote the variant-specific values for variant $i$ by $g_i$, $d_{avi}$, and so on.

### Respiration rates

We also measured animal respiration rates $m(t)$ and included these in our model. The amount of infectious virus an animal deposits into the air per unit time is $m(t) V_a(t)$ and the amount of sgRNA the animal deposits per unit time is $m(t) N_a(t)$.

### Sex effects

Since male hamsters are physically larger and appeared to have different shedding profiles, we wanted to be able to estimate the effect of host sex on key parameters of interest. To do this, we modeled males as possibly offset from females in their typical values of respiration rate $m(t)$, airborne shedding exponential growth rate $g$, airborne shedding exponential decay rate $d_{av}$, peak airborne shedding time $t_a$, and peak airborne shedding rate $V_{max}$ (this then has downstream consequences for other parameters such as $d_{an}$ or $t_o$ that depend on those core virological parameters).

We modeled sex differences in the virological parameters via offsets to the mean log values for male hamsters. $\Delta_x$ denotes the offset for variable $x$. So for example if females have a mean log respiration rate of $\log[m]$, males have one of $\log[m] + \Delta_m$. We also estimated male offsets $\Delta_g$ for growth rate $g$, $\Delta_d$ for decay rate $d_{av}$, and $\Delta_V$ for peak shedding $V_{max}$. We did not treat effects as variant-specific, but rather sought to estimate the average sex differences in infection dynamics across the two variants tested.

### Individual heterogeneity in disease course

To account for the fact that individuals have heterogeneous disease courses, we made our model hierarchical, with core virological parameter values for specific individuals distributed about the typical population values. If infected with a variant $i$, animal $j$ has individual values for the virus growth rate $g_{ij}$, the virus decay rate $d_{avij}$, the peak load time $t_{aij}$, and the peak load $V_{max_{ij}}$.

These values are log-normally distributed about the population values for the given variant and animal sex, with estimated variant-specific standard deviations $\sigma_{gi}$, $\sigma_{di}$, $\sigma_{ti}$, and $\sigma_{Vi}$. We use $s_j$ as an indicator for the sex of hamster $j$ (0 if female, 1 if male). Then:

$$
\begin{aligned}
\log[g_{ij}] &\sim \text{Normal}\left(\log[g_i] + s_j\Delta_g, \sigma_{gi}\right) \\
\log[d_{avij}] &\sim \text{Normal}\left(\log[d_{avi}] + s_j\Delta_d, \sigma_{di}\right) \\
\log[t_{aij}] &\sim \text{Normal}\left(\log[t_{ai}], \sigma_{ti}\right) \\
\log\left[V_{max_{ij}}\right] &\sim \text{Normal}\left(\log\left[V_{max_i}\right] + s_j\Delta_V, \sigma_{Vi}\right)
\end{aligned}
\tag{11}
$$

We also allowed for individual heterogeneity in respiration rates: animal $j$ has an individual time-averaged respiration rate $m_j$ log-normally distributed about the population value $m$ with an estimated standard deviation $\sigma_m$:

$$\log[m_j] \sim \text{Normal}\left(\log[m] + s_j\Delta_m, \sigma_m\right) \tag{12}$$

## Predicting observable quantities

We measured the following quantities:

- Virus subgenomic RNA (sgRNA) in oral swabs, measured via quantitative PCR (qPCR) in units of estimated copy numbers.

- Infectious virus in oral swabs, measured via endpoint titration in units of $\log_{10} TCID_{50}$ per mL.
- Respiration rates, measured via plethysmography in units of mL air exhaled per unit time.
- Virus sgRNA collected on cage air filters over 24 hr sampling periods, measured via qPCR in units of estimated copy numbers.
- Infectious virus collected on cage air filters over 24 hr sampling periods, measured via plaque assay as total plaques formed.
- Infection statuses for each variant for each sentinel hamster.

## Units of virus dynamics

We expressed $V_a(t)$ and $V_o(t)$ in units of total filter-collectible plaque forming units (PFU) shed per mL/h (i.e. units that directly predict the cage air infectious virus measurements).

As discussed in section 2, our model explicitly relates infectious virus dynamics $V_a(t)$ and $V_o(t)$ to sgRNA copy number dynamics $N_a(t)$ and $N_o(t)$. The distinct conversion factors $a_n$ and $o_n$ and decay rates $d_{an}$ and $d_{on}$ for airborne versus oral shedding account for two types of possible differences between airborne and oral samples: biological differences (distinct underlying relationships between infectious virus concentration and sgRNA concentration) and measurement differences (distinct quantities of absolute sgRNA quantity recovered given the same underlying sgRNA concentration).

## Converting predicted swab virus to units of $TCID_{50}$

To fit our model, we needed to convert our internal representation of predicted oral shedding of virus $V_o(t)$, which has the same 'predicted air plaques' units as airborne shed virus $V_a(t)$, into the units in which we measured oral shedding: infectious virus $v_o(t)$ in units of $\log_{10} TCID_{50}$ mL. We modeled this conversion via a multiplier $o_v$:

$$v_o(t) = o_v V_o(t) \tag{13}$$

The multiplier $o_v$ subsumes both unit conversion and any actual multiplicative difference in virion numbers between airborne shedding and swabs (which could come from lower peak virion concentrations, sampling volume, et cetera.).

## Predicting air sample plaques

We measured shedding into the air at the cage level; multiple hamsters were housed within a single cage. Moreover, samples were cumulative 24 hr accumulation on the air filter, rather than a point-sample.

### Cumulative shedding over a time period

So to fit our model, we needed to compute the cumulative airborne shedding $A$ over some time period $(t_1, t_2)$:

$$A(t_1, t_2) = \int_{t_1}^{t_2} m(t)\, V_a(t)\, dt \tag{14}$$

where $m(t)$ is the animal's respiration rate. Integrating yields:

$$A(t_1, t_2) = \begin{cases} \dfrac{m}{g}\left(V_a(t_2) - V_a(t_1)\right) & t_2 \leq t_a \\[2mm] \dfrac{m}{d_{av}}\left(V_a(t_1) - V_a(t_2)\right) & t_1 > t_a \\[2mm] \dfrac{m}{g}\left(V_a(t_a) - V_a(t_1)\right) + \dfrac{m}{d_{av}}\left(V_a(t_a) - V_a(t_2)\right) & otherwise \end{cases} \tag{15}$$

where $m$ is an appropriately-chosen constant to represent the time-varying effect of $m(t)$ on the value of the integral. Note that while ideally, we would know how $m(t)$ and $V_a(t)$ change together and compute the integral explicitly, in practice we could only measure $m(t)$ coarsely, and so it was simpler to infer the appropriate value, understanding that it would not necessarily equal a naive temporal average.

### Accounting for loss of virion infectivity

Furthermore, since the air shedding measured accumulation on the air filter over a 24 hr period, we had to account for decay of infectious virus between exhalation and quantification. To do this, we assumed that the virus decays exponentially in suspended aerosols and on the filter (as we have

previously measured empirically *Morris et al., 2021*; *van Doremalen et al., 2020*) but that minimal virus is lost once the filter is removed for virus quantification.

Suppose that the sampling period begins at a time $t_1$ post-infection and ends at a time $t_2$ when the filter is removed. Each hamster $j$ sheds infectious virus at a rate $mV_{aj}(t)$ per unit time. But if the virus loses infectivity according to an exponential decay process with rate $\lambda$, then only a fraction $e^{-\lambda(t_s-t)}$ of the virions shed at time $t_s > t_1$ and collected on the filter will remain infectious when the filter is collected at $t_2$. For simplicity in mathematical notation, we assume here that, as in our experiments, potentially shedding individual remained present until at least $t_2$, though our code allows for modeling other situations, such as the removal of the shedding individual prior to the removal of the filter.

We denote the cumulative number of virions shed since $t_1 \geq t_0$ that remain viable (infectious) at $t_2 \geq t_1$ by $A_v(t_1, t_2)$ (to distinguish it from the cumulative shedding irrespective of retained infectiousness $A$). Fixing $t_1$, $A_v$ is defined by the following differential equation with respect to $t_2$:

$$\frac{dA_v}{dt_2} = mV_a(t_2) - \lambda A_v \tag{16}$$

Since we have an expression for $V_a(t)$ consisting of fixed periods of constant-rate exponential growth or decay, we can treat this as a system of two coupled linear ordinary differential equations, with $\frac{dV_a}{dt_2} = gV_a$ or $\frac{dV_a}{dt_2} = -d_{av}V_a$, depending on whether $t_2 > t_a$, and solve.

Assume the shedding individual is removed at a time $t_s \geq t_1$, which may or may not be after $t_a$. Then:

$$A_v(t_1, t_2) = \begin{cases} \frac{mV(t_1)}{g+\lambda}\left(e^{g(t_2-t_1)} - e^{-\lambda(t_2-t_1)}\right) & t_1 < t_2 \leq t_a, t_s \\ \frac{mV(t_1)}{-d_{av}+\lambda}\left(e^{-d_{av}(t_2-t_1)} - e^{-\lambda(t_2-t_1)}\right) & t_a \leq t_1 < t_2 \leq t_s \\ \frac{mV(t_a)}{-d_{av}+\lambda}\left(e^{-d_{av}(t_2-t_a)} - e^{-\lambda(t_2-t_a)}\right) + A_v(t_1, t_a)e^{-\lambda(t_2-t_a)} & t_1 < t_a < t_2 \leq t_s \\ A_v(t_1, t_s)e^{-\lambda(t_2-t_s)} & t_1 \leq t_s < t_2 \end{cases} \tag{17}$$

In a previous version of this work, we took an equivalent alternative approach based on integrating over the virions shed at a time $t_1 \leq t \leq t_2$ that would survive until $t_2$; we describe that alternative approach in section 8.2 below for completeness.

## Predicting air sample sgRNA

For air sample sgRNA, we again predicted cumulative accumulation. We did not model environmental degradation of detectable sgRNA, but rather chose to treat it implicitly via the decay rate ratio $q_n$ relating airborne infectious virus shedding to airborne sgRNA shedding. We chose not to model sgRNA environmental degradation more explicitly because environmental half-lives for sgRNA are less well-characterized, but likely longer, than environmental half-lives for infectious virus.

The cumulative predicted number of sgRNA copies collected is:

$$A_n(t_1, t_2) = m\int_{t_1}^{t_2} N_a(t)\,dt \tag{18}$$

$A_n$ can be computed using the following antiderivative:

$$G(t, a, t_i) = \int \exp\left[a(t-t_i)\right]dt = \frac{1}{a}\exp\left[a(t-t_i)\right] \tag{19}$$

We obtain:

$$A_n(t_1, t_2) = \begin{cases} mN_a(t_1)\left[G(t_2, g, t_1) - G(t_1, g, t_1)\right] & t_2 < t_a \\ mN_a(t_1)[G(t_2, -d, t_1) - G(t_1, -d, t_1)] & t_1 > t_a \\ mN_a(t_1)[G(t_a, g, t_1) - G(t_1, g, t_1)] + \\ mN_a(t_a)[G(t_2, -d, t_a) - G(t_a, -d, t_a)] & t_1 \leq t_a \leq t_2 \end{cases} \tag{20}$$

## Predicting sentinel exposures

Using our kinetics model, we were able to estimate the probability each donor in our dual donor experiment had of infecting each sentinel, taking into account donor sex, infecting variant, and timing of exposure. This also enabled us to assess whether the absence of observed co-infections in sequential donor experiments was more suggestive of competitive interference or non-interference among the virus variants (see section Assessing coinfection probabilities for methods and results).

To do this, we assumed that each sentinel's dose from each donor was proportional to the cumulative airborne shedding by the donor over the period of sentinel exposure. Given the short exposure period, we ignored the effect of environmental loss, so the computation was $A_v(t_1, t_2)$ as in *Equation 17*, but with the environmental loss rate set to $\lambda = 0$. We assumed that the total dose received by each sentinel was equal to the cumulative shedding multiplied by an estimated variant-specific constant $c_i$ that subsumes uncertainty about sentinel respiration rate, cage airflow, and per-virion infectivity when inhaled by a hamster (since $A_v$ has units of predicted *cell culture* plaques, and hamster airways may be more or less susceptible to virions of a given variant).

So in our model, each sentinel $j$ receives a dose $h_{ij}$ of variant $i$ that depends on the virus shedding $A_{vij}(t_{1ij}, t_{2ij})$ from the donor animal associated to variant $i$ and sentinel $j$, where $t_{0ij}$ and $t_{1ij}$ are the start and end times of the exposure under the exposure design for variant $i$ and sentinel $j$:

$$h_{ij} = c_i A_{vij}(t_{1ij}, t_{2ij}) \tag{21}$$

We again applied a Poisson single-hit model of infection, so the probability $p_{inf}(i, j)$ that sentinel $j$ is infected with variant $i$ depends on the cumulative dose $h_{ij}$ as:

$$p_{inf}(i, j) = 1 - e^{-h_{ij}} \tag{22}$$

## Relating predicted quantities to observed quantities
## Oral swabs
### Infectious virus titers

Denote the $k$th measured oral swab titer by $y_{vok}$. Suppose it was sampled from individual animal $j$ at time $t$. Then its predicted value is $v_{ok} = o_v V_{oj}(t)$.

We modeled the distribution of observed swab titers given $\log_{10} TCID_{50}$ values $v_{ok}$ via a Poisson single-hit process, as we have described previously *Gamble et al., 2021*; *Morris et al., 2021* (in particular, see section Offset growth and decay of oral shedding of SI of *Gamble et al., 2021*). Briefly, in a Poisson single-hit model of virus titration, a Poisson-distributed number of virions successfully infect cells in each. The mean µ of this Poisson depends on the underlying sample virus concentration $v$ (in $\log_{10} TCID_{50}$) and degree of dilution $D$ (in $\log_{10}$ fold dilutions).

$$\mu = \log(2) \, 10^{v-D} \tag{23}$$

The factor of $\log(2)$ converts from units of $TCID_{50}$ to units of successful virions.

A well will be positive for infection if at least one virion infects a cell, which occurs with probability:

$$1 - \exp\left(-\log(2) \, 10^{v-D}\right) \tag{24}$$

A complication to the typical single-hit model in this case is that we only had total counts of positive wells rather than exact well identities, dilutions, and positive/negative status. To handle this, we used an approximate method that integrates the likelihood function over the most probable configurations of positive and negative wells that could generate an observed total count. We describe this method in section Well observation process .

### Subgenomic RNA

If $y_{nok}$ is the $k^{th}$ measurement of oral swab sgRNA, sampled from animal $j$ at time $t$, its predicted value is $n_{ok} = N_o(t)$. To account for different sampling procedures and qPCR runs for the donor animals used in the dual donor experiments compared to the animals used in the kinetics experiments, for the donor animals we added an estimated offset term $f$ to the log copy number: $\log[n_{ok}] = \log[N_o(t)] + f$.

We modeled the observed $\log_{10}$ oral swab sgRNA copy numbers $y_{nok}$ as distributed about their predicted values $n_{ok}$, with an estimated variant-specific standard deviation $\sigma_{noi}$ (where $i$ is the variant infecting animal $j$) and censoring at the minimum and maximum observable values (which are given by the particular sgRNA standard curve).

$$\log_{10}\left(y_{noj}\right) \sim \text{CensoredNormal}\left(\log_{10}\left(n_{oj}\right), \sigma_{noi}, n_{min}, n_{max}\right) \tag{25}$$

## Air samples

### Plaques

We used **Equation 17** to predict the number of plaques $v_{ak}$ observed on each filter.

Note that this implies $V_a\left(t\right)$ has units of filter plaques produced per mL exhaled air per unit time (in the absence of environmental decay).

If an observed plaque count $y_{avk}$ comes from an air sample taken between time $t_1$ and time $t_2$ from a cage with $n_h$ hamsters infected with variant $j$, the corresponding predicted plaque count $v_{ak}$ is:

$$v_{ak} = \sum_{u=1}^{n_h} A_{vu}\left(t_1, t_2\right) \tag{26}$$

where $A_{vu}\left(t_1, t_2\right)$ is $A_v\left(t_1, t_2\right)$ for the $u$th hamster.

Since in vitro cell infection is well-described by a Poisson single-hit process (**Brownie et al., 2011**) (possibly with binomial thinning), we modeled the observed plaque counts $y_{avk}$ as Poisson-distributed about their predicted values $v_{ak}$.

$$y_{vak} \sim \text{Poisson}\left(v_{ak}\right) \tag{27}$$

### Subgenomic RNA

Similarly, we used **Equation 20** to predict the number of sgRNA copies $n_{ak}$ that would be observed when sampling cage $k$ from $t_1$ until $t_2$ as:

$$n_{ak} = \sum_{u=1}^{n_h} A_{nu}\left(t_1, t_2\right) \tag{28}$$

where $A_{nu}$ is the cumulative sgRNA shedding function $A_n$ for hamster $u$.

We model the observed $\log_{10}$ air sample sgRNA copy numbers $\log_{10}\left(y_{nak}\right)$ as normally distributed about their predicted values $\log_{10}\left(n_{ak}\right)$ with an estimated variant-specific standard deviation $\sigma_{nai}$ and censoring at the minimum and maximum possible $\log_{10}$ estimated copy numbers (which depend on the standard curve).

$$\log_{10}\left(y_{nak}\right) \sim \text{CensoredNormal}\left(\log_{10}\left[n_{ak}\right], \sigma_{nai}, n_{min}, n_{max}\right) \tag{29}$$

### Respiration rates

We modeled the observed log respiration rates for animal $j$ $\log\left(y_{mij}\right)$ as distributed about the animal's typical log value $\log\left(m_j\right)$ with an estimated standard deviation $\sigma_r$.

$$\log\left(y_{mij}\right) \sim \text{Normal}\left(\log\left(m_j\right), \sigma_r\right) \tag{30}$$

### Sentinel infection status

Our dynamical model generates predicted infection probabilities $p_{inf}\left(i,j\right)$ for sentinel $i$ with variant $j$ (see section Predicting sentinel exposures).

The observed infection status for sentinel $j$ with variant $i$, $y_{pij} \in \{0,1\}$ is therefore Bernoulli distributed with probability $p_{inf}\left[i,j\right]$.

$$y_{pij} \sim \text{Bernoulli}\left(p_{inf}\left[i,j\right]\right) \tag{31}$$

## Prior distributions

In general, we sought to set prior distributions for our parameters that were "weakly informative" (*Gelman et al., 2008*); that is, that rule out biologically implausible or impossible values while remaining fairly agnostic about possible values of interest. We assessed the robustness of our prior distribution choices via prior predictive checks.

### Respiration rates

We placed a normal prior on the population-wide mean log respiration rate $\log(m)$, with $m$ in units of mL h$^{-1}$.

$$\log(m) \sim \text{Normal}(\log[4800], 0.25) \tag{32}$$

We placed a positive-constrained normal prior on the individual respiration rate standard deviation $\sigma_{mi}$ (see section Individual heterogeneity in disease course).

$$\sigma_{mi} \sim \text{PosNormal}(0, 0.25) \tag{33}$$

### Virological parameters

We placed log-normal priors on the variant-specific time to peak $t_{ai}$ and peak shedding rate $V_{max_i}$; $i$ indexes the variant. To encode prior information about the variant-specific growth and decay rates $g_i$ and $d_{avi}$ in an interpretable manner, we placed normal priors on the doubling and halving times (in hours) $t_{2i} = \log(2)/g_i$ and $t_{\frac{1}{2}i} = \log(2)/d_{avi}$ and then back-calculated $g_i$ and $d_{avi}$.

$$
\begin{aligned}
\log[t_{ai}] &\sim \text{Normal}(\log[24], 0.5) \\
\log[V_{max_i}] &\sim \text{Normal}(\log[1] - \log[24] - \log[4800], 3) \\
\log[t_{2i}] &\sim \text{Normal}(\log[5], 0.5) \\
\log\left[t_{\frac{1}{2}i}\right] &\sim \text{Normal}(\log[15], 0.75) \\
\log\left[t_{\frac{1}{2}i}\right] &\sim \text{Normal}(\log[15], 0.75)
\end{aligned}
\tag{34}
$$

The prior mode for $\log[V_{max_i}]$ can be interpreted as corresponding to the amount of shedding that would lead to 1 plaque(s) on the air filter from a 24 h sample at the prior mean respiration rate of $\log[4800\,\text{mL/h}]$.

We placed normal priors on the male sex effects $\Delta_m$, $\Delta_g$, $\Delta_d$, and $\Delta_V$ that modify the virological parameters.

$$
\begin{aligned}
\Delta_m &\sim \text{Normal}(0, 0.25) \\
\Delta_g &\sim \text{Normal}(0, 0.25) \\
\Delta_d &\sim \text{Normal}(0, 0.25) \\
\Delta_V &\sim \text{Normal}(0, 0.25)
\end{aligned}
\tag{35}
$$

We placed lognormal priors on the swab to air peak timing ratio $\omega$, the swab to air decay rate ratio $q_o$, the sgRNA to infectious virus decay rate ratio $q_n$, the air infectious virus to oral TCID conversion factor $o_v$, the air and swab copy number to infectious virus ratios $a_n$ and $o_n$. We placed a normal prior on the donor log copy number offset $f$.

$$
\begin{aligned}
\log(\omega) &\sim \text{Normal}(0, 0.25) \\
\log(q_o) &\sim \text{Normal}(0, 1) \\
\log(q_n) &\sim \text{Normal}(\log[1], 1) \\
\log(o_v) &\sim \text{Normal}(0, 10) \\
\log(a_n) &\sim \text{Normal}(0, 10) \\
\log(o_n) &\sim \text{Normal}(0, 10) \\
f &\sim \text{Normal}(0, 1.5)
\end{aligned}
\tag{36}
$$

We placed positive-constrained normal priors on the hierarchical standard deviations that specify degree of individual variation about these population-wide virological parameters (see section Individual heterogeneity in disease course).

$$
\begin{aligned}
\sigma_{gi} &\sim \text{PosNormal}(0, 0.2) \\
\sigma_{di} &\sim \text{PosNormal}(0, 0.2) \\
\sigma_{ti} &\sim \text{PosNormal}(0, 0.15) \\
\sigma_{Vi} &\sim \text{PosNormal}(0, 2)
\end{aligned}
\tag{37}
$$

### Sentinel infection process constant
We placed a lognormal prior on the variant-specific sentinel infection process constant $c_i$.

$$
\log[c_i] \sim \text{Normal}(0, 3)
\tag{38}
$$

### Observation error standard deviations
We placed positive-constrained normal priors on the observation process standard deviations for respiration rate $\sigma_r$, oral swab sgRNA copies $\sigma_{noi}$, and air sample sgRNA copies $\sigma_{nai}$.

$$
\begin{aligned}
\sigma_r &\sim \text{PosNormal}(0, 0.2) \\
\sigma_{noi} &\sim \text{PosNormal}(0, 0.5) \\
\sigma_{nai} &\sim \text{PosNormal}(0, 0.5)
\end{aligned}
\tag{39}
$$

### Prior predictive checks
We assessed the appropriateness of prior choices via prior predictive checks. *Appendix 1—figure 1* shows a version of main text *Figure 1* but where sample trajectories are plotted alongside the data but drawn from the prior predictive distribution rather than the posterior distribution.

### Assessing coinfection probabilities
To assess the probability of coinfection, we visualized the infection probabilities for each variant in each cage. Given no interaction between two variants' infection processes, the probability of being coinfected for each hamster $j$ is the product of the hamster's probabilities for each variant:

$$
P_{coinf}(j) = P_{inf}(1, j) P_f(2, j)
\tag{40}
$$

The distribution of the number of coinfections in a given cage or experiment is then the convolution of these individual Bernoulli-distributed outcomes for individuals.

### Results
*Appendix 1—figure 2* shows the estimated infection probabilities by variant and cage. Cage F was the only cage in which we observed coinfections, and our model shows that it is indeed the only cage in which both Alpha and Delta clearly had a high probability of causing infections in the sentinels.

We then calculated the posterior estimated probability of coinfection occurring for each hamster in each cage, according to *equation (40)*. *Appendix 1—figure 3* shows the resulting estimates.

The model estimates that coinfection probabilities were highest in Cage F simply because both Alpha and Delta infection probabilities were high. In other cages, coinfection probabilities are substantially lower, since at least one variant has a low individual infection probability 2. Cage C

(Delta, then Alpha) is the only sequential exposure cage in which the absence of coinfections is perhaps surprising; even there, the data are consistent with a coinfection probability of under 25% or even under 10%, so given that only 5 sentinels were exposed, the absence of a coinfection is consistent with random variation.

Finally, to assess whether the absence of any coinfections in the sequential experiments while several were observed in the simultaneous experiments could be explained by chance, we calculated the posterior distribution for the expected number of coinfections by experiment type (this is the sum of the probability for each cage times the number of sentinels in that cage). The results are shown in *Appendix 1—figure 4*.

The model suggests that the absence of coinfections in the sequential exposures could simply result from low probabilities in all cages except C; the data are consistent with an expectation of between zero and two coinfections, though more than that would also have been plausible.

Taken together, our results suggest that differences in donor virus dynamics and shedding could readily explain the differences between sequential and simultaneous exposures in our small-N dataset. Identifying or ruling out competitive (or facilitating) interaction among virus variants during sequential versus simultaneous transmission would likely require larger samples.

## Computational methods

We implemented and conducted inference from our model in Python using the Numpyro probabilistic programming framework (*Phan et al., 2019*). We drew posterior samples using Numpyro's iterative implementation (*Phan et al., 2019*) of the No-U-Turn Sampler (NUTS) (*Hoffman and Gelman, 2014*), a form of Hamiltonian Monte Carlo (HMC). For each model fit, we ran three Markov Chains, each with 1000 warmup steps and 1000 samples. We assessed convergence by examining figures and by confirming sufficient effective sample sizes and the absence of divergent transitions after warmup.

For inference purposes, we set a minimum rate of for all Poisson distributions of "hitting" virions (for filter plaques and virus titration). True rates can cause numerical issue when conducting NUTS sampling with Numpyro. This minimum rate of can be thought of as representing a very small temperature and humidity on inactivation of SARS-CoV-2 and other enveloped vprobability of a false positive plaque or well.

We prepared data for modeling and analyzed and visualized output in Python *Van Rossum and Drake, 2009*; the packages Numpy (*Harris et al., 2020*), Scipy (*Jones et al., 2001*), Matplotlib (*Hunter, 2007*), and Polars (*Vink, 2022*) were particularly critical.

All code and data to reproduce Bayesian inference results, including model fits and model output figures, is available on the project Github repository (https://github.com/dylanhmorris/host-viral-determinants) and archived on Zenodo (https://doi.org/10.5281/zenodo.8396135).

## Additional mathematical details

### Well observation process

Spearman-Karber estimates for a 0.1 mL inoculum give an exact value for the total number of positive wells in a 4 well by 8 serial dilution series (with a an undilute first row , and serial -fold dilutions—here —such that the row has been diluted -fold). This allowed us to back-calculate the total number of positive wells. With some additional assumptions, we were then able to calculate the approximate likelihood of observing a given number of positive wells given a true underlying titer .

We assumed that wells were all positive until some dilution and wells were all negative or not attempted at dilutions and higher. That is, only two sequential dilutions are assumed potentially to have a mix of positive and negative wells: and . This is by far the most probable way to produce a given total number of positive wells, particularly with 10-fold or coarser dilutions, since other hit distributions require some negative wells to occur at substantially higher plated concentrations than some positive wells.

With this assumption made, a given total number of positive wells has implies a value of this and a corresponding number of total positives seen at dilutions $d$ and $d$+1 combined:

$$
\begin{aligned}
k &= \min\{n, 4 + n \mod 4\} \\
d &= \frac{n - k}{4}
\end{aligned}
\tag{41}
$$

where mod denotes the modulo operation (remainder when n is divided by 4). Note that while we assume $d$ and $d+1$ can have a mix of positive wells, we do not assume for certain that they do. When $k=4$, all positive at and all negative at $d+1$ is a possible outcome (or, much less probably, all positive at $d+1$ and all negative at $d$); it is just not the *only* possible outcome, as we could also have, e.g., 3 of 4 wells positive at and 1 of 4 positive at $d+1$.

The approximate log likelihood for observing $n$ positive wells given an underlying virus concentration $v$ is then the sum over the possible ways to generate $k$ positives at dilutions $d$ and $d+1$. Define the random variables $K_d$ and $K_{d+1}$ as the number of wells positive at dilutions $d$ and $d+1$, respectively. Then the probability of observing a certain value of $K_d$ given the underlying virus concentration $v$ is given by a binomial distribution with success probability equal to the single hit probability at dilution $d$, i.e.:

$$P\left(K_d = k \lor v\right) = \binom{4}{k} p^k \left(1 - p\right)^{4-k} \tag{42}$$

For our experiments, $v$ is measured in $log_{10}TCID_{50}$ and dilutions are 10-fold, so we have

$$p = 1 - \exp\left(-\log\left(2\right) 10^{v-d}\right)$$

This gives us a computable approximate likelihood $L\left(n \lor v\right)$ :

$$L\left(n \lor v\right) \approx \sum_{c=k-4}^{4} \log\left[P\left(K_d = c \lor v\right)\right] + \log\left[P\left(K_{d+1} = k - c \lor v\right)\right] \tag{43}$$

## Alternative approach to computing viable virions shed during a time window

This describes a second, equivalent manner of computing $A_v\left(t_1, t_2\right)$ , the cumulative number of virions shed between $t_1$ and $t_2$ that remain viable at $t_2$ . In this instance, we integrate over the time of shedding t, but thin the virions shed by considering only the fraction $e^{-\lambda\left(t_2-t\right)}$ that will be viable when sampled at $t_2 \geq t$. This fraction is exact if we treat virion loss of infectivity as exponential at a rate $\lambda$ .

$$A_v\left(t_1, t_2\right) = m \int_{t_1}^{t_2} V_a\left(t\right) e^{-\lambda\left(t_2-t\right)} dt \tag{44}$$

This integral can be computed using the following antiderivative:

$$F\left(t, a, t_i, t_s\right) = \int \exp\left[a\left(t - t_i\right) - \lambda\left(t_s - t\right)\right] dt = \frac{1}{a + \lambda} \exp\left[a\left(t - t_i\right) - \lambda\left(t_s - t\right)\right] \tag{45}$$

In our case, $a$ is the exponential growth rate of shedding, with negative $a$ values representing exponential decay, and $t_s \geq t$ is the time of filter removal (so $t_s - t$ is how long a virion deposited at $t$ must retain infectiousness in order to be infectious when the filter is removed).

To compute $A_v\left(t_1, t_2\right)$ , we have to consider several possible cases. Recall that time is measured relative to the time $t = 0$ that the shedding individuals were inoculated, and $t_a > 0$ is the time of peak air shedding.

If $t_2 < t_a$ , the entire sampling period happens before peak air shedding. In that case:

$$A_v\left(t_1, t_2\right) = mV_a\left(t_1\right)\left[F\left(t_2, g, t_1, t_2\right) - F\left(t_1, g, t_1, t_2\right)\right] \tag{46}$$

If $t_1 > t_a$ , the entire sample is taken after air shedding has peaked. In that case:

$$A_v\left(t_1, t_2\right) = mV_a\left(t_1\right)\left[F\left(t_2, -d_{av}, t_1, t_2\right) - F\left(t_1, -d_{av}, t_1, t_2\right)\right] \tag{47}$$

If the air shedding peak occurs during sampling ($t_1 \leq t_a \leq t_2$), the problem can be solved piecewise:

$$A_v\left(t_1, t_2\right) = \int_{t_1}^{t_a} mV_a\left(t_1\right)\exp\left[g\left(t - t_1\right)\lambda\left(t_2 - t\right)\right]dt +$$
$$\int_{t_a}^{t_2} mV_a\left(t_a\right)\exp\left[-d_{av}\left(t - t_a\right) - \lambda\left(t_2 - t\right)\right]dt \qquad (48)$$

And so applying the antiderivative $F$ from *Pitol and Julian, 2021*:

$$A_v\left(t_1, t_2\right) = mV_a\left(t_1\right)\left[F\left(t_a, g, t_1, t_2\right) - F\left(t_1, g, t_1, t_2\right)\right] +$$
$$mV_a\left(t_a\right)\left[F\left(t_2, -d_{av}, t_a, t_2\right) - F\left(t_a, -d_{av}, t_a, t_2\right)\right] \qquad (49)$$

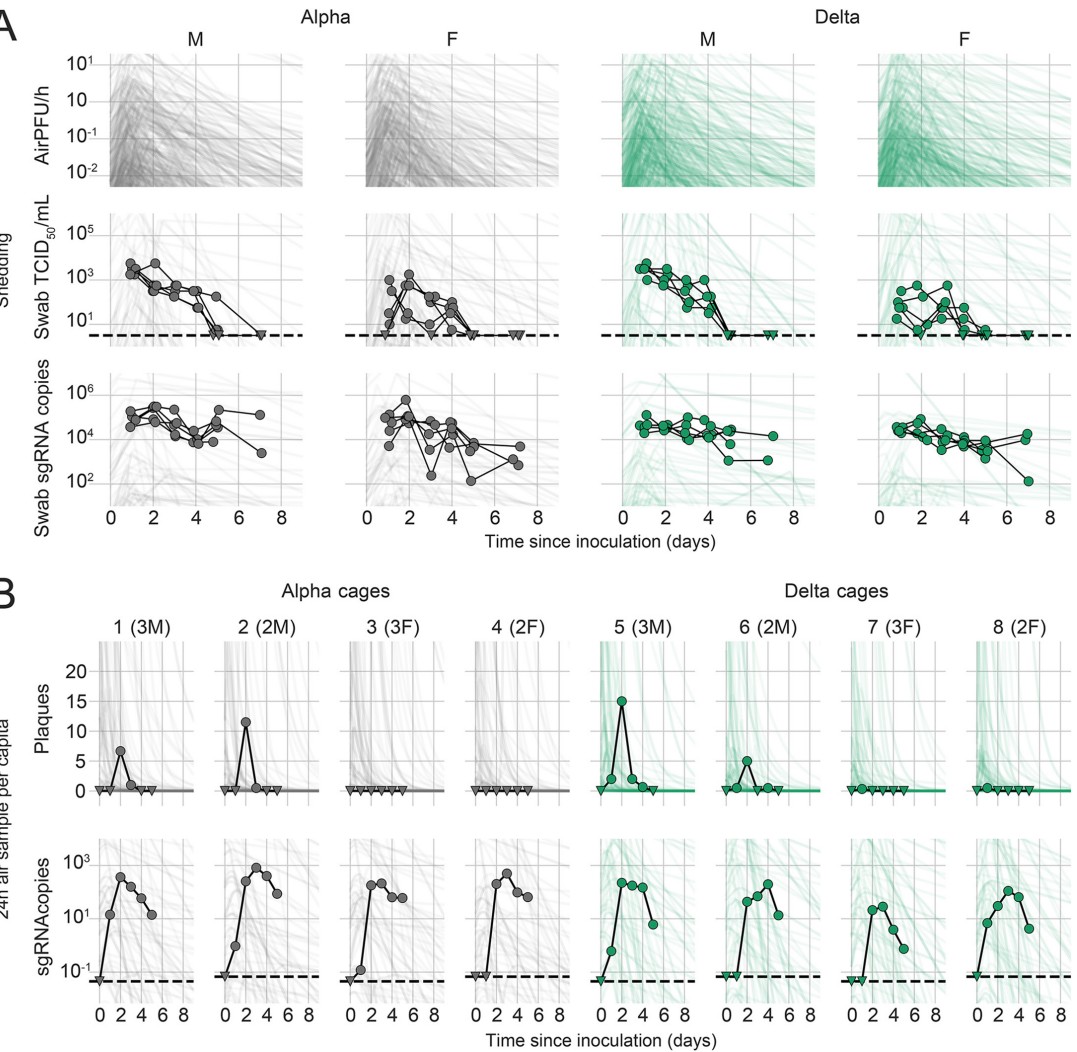

**Appendix 1—figure 1.** Prior predictive checks. Figure is as main text *Figure 1*, except that the trajectories showed by the semi-transparent lines are randomly sampled from the prior predictive distribution rather than from the posterior, and 500 lines are drawn in panel A rather than 100, given the greater dispersion relative to the data. Wide range of simulated trajectories relative to the data shows that priors allow for a wide range of a priori plausible kinetics.

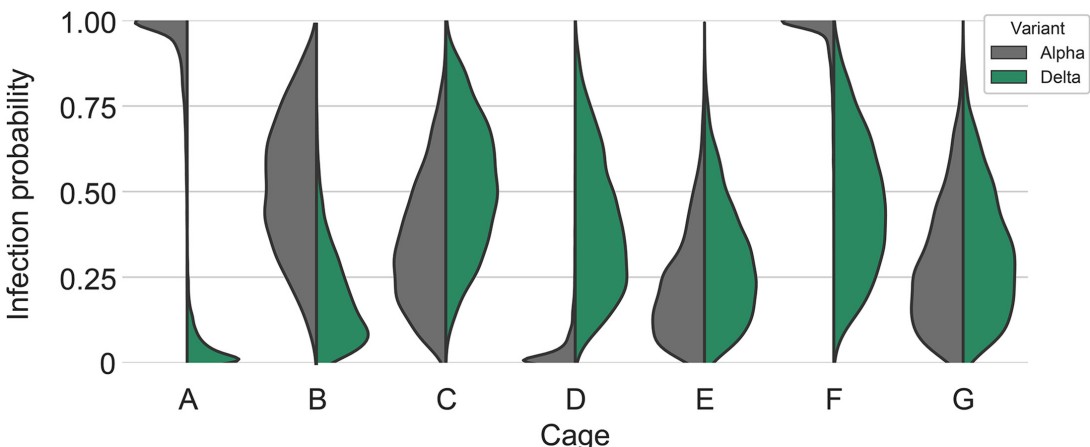

**Appendix 1—figure 2.** Posterior estimates for the infection probabilities for each variant in each cage. Half-violin plots show posterior densities for Alpha infection probability (gray) compared to Delta infection probability (green). There were very high Alpha infection probabilities in cages **A** and **F**.

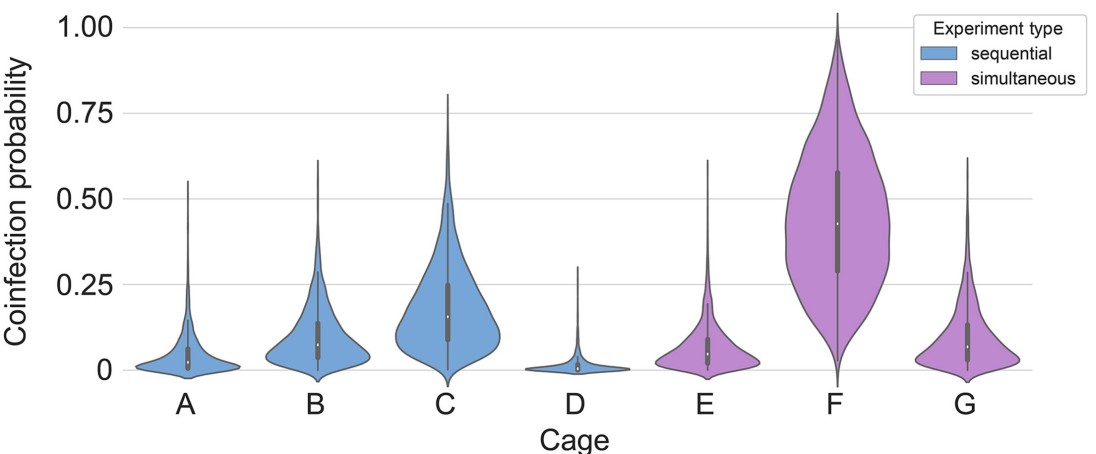

**Appendix 1—figure 3.** Posterior estimates for coinfection probabilities by cage. Sequential exposures shown in blue, simultaneous exposures shown in pink. Cage F, where coinfections were actually observed, has a substantially higher coinfection probability estimate than other cages.

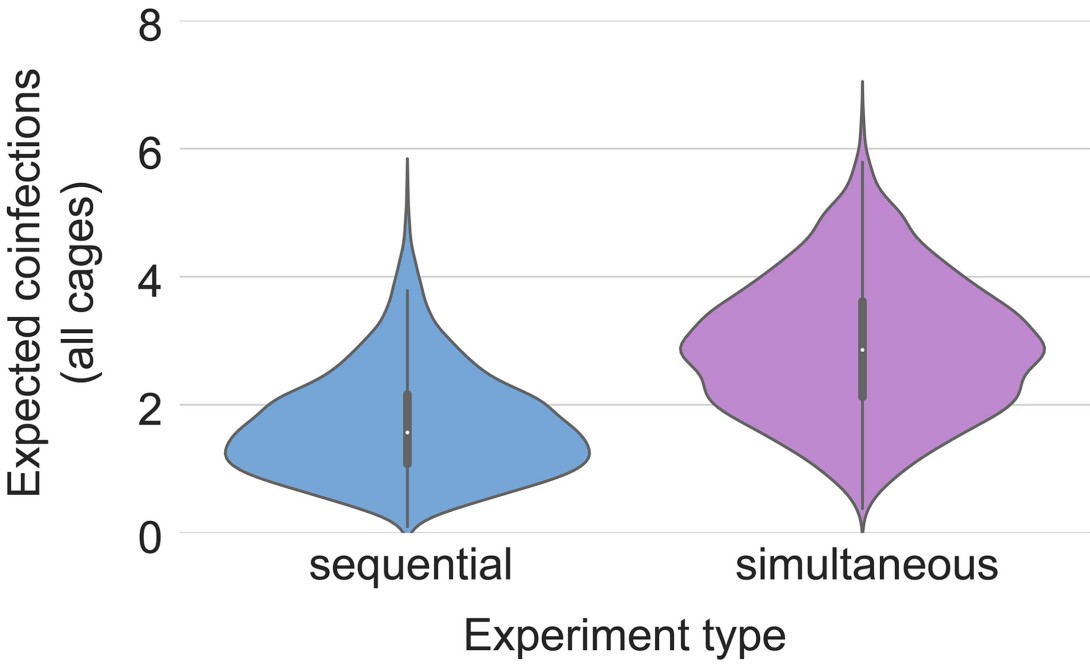

**Appendix 1—figure 4.** Expected coinfection counts for sequential and simultaneous experiments.

