## [Editor Report · eLife assessment]

This manuscript describes rigorous experiments that provide a wealth of virologic, respiratory physiology, and particle aerodynamic data pertaining to aerosol transmission of SARS-CoV-2 between infected Syrian hamsters. The significance of the paper is **fundamental** because infection is compared between alpha and delta variants, and because viral load is assessed via numerous assays (gRNA, sgRNA, TCID) and in tissues as well as the ambient environment of the cage. The strength of evidence is **compelling**.

---

## [Referee Report · Reviewer #1 (Public Review)]

In the submitted manuscript, Port et al. investigated the host and viral factors influencing the airborne transmission of SARS-CoV-2 Alpha and Delta variants of concern (VOC) using a Syrian hamster model. The authors analyzed the viral load profiles of the animal respiratory tracts and air samples from cages by quantifying gRNA, sgRNA, and infectious virus titers. They also assessed the breathing patterns, exhaled aerosol aerodynamic profile, and size distribution of airborne particles after SARS-CoV-2 Alpha and Delta infections. The data showed that male sex was associated with increased viral replication and virus shedding in the air. The relationship between co-infection with VOCs and the exposure pattern/timeframe was also tested. This study appears to be an expansion of a previous report (Port et al., 2022, Nature Microbiology). The experimental designs were rigorous, and the data were solid. These results will contribute to the understanding of the roles of host and virus factors in the airborne transmission of SARS-CoV-2 VOCs.

---

## [Referee Report · Reviewer #2 (Public Review)]

This manuscript by Port and colleagues describes rigorous experiments that provide a wealth of virologic, respiratory physiology, and particle aerodynamic data pertaining to aerosol transmission of SARS-CoV-2 between infected Syrian hamsters. The data is particularly significant because infection is compared between alpha and delta variants, and because viral load is assessed via numerous assays (gRNA, sgRNA, TCID) and in tissues as well as the ambient environment of the cage. The paper will be of interest to a broad range of scientists including infectious diseases physicians, virologists, immunologists and potentially epidemiologists.

---

## [Author Response]

The following is the authors’ response to the original reviews.

eLife assessmentThis paper investigates host and viral factors influencing transmission of alpha and delta SARS-CoV-2 variants in the Syrian hamster model and fundamentally increases knowledge regarding transmission of the virus via the aerosol route. The strength of evidence is solid and could be improved with a clearer presentation of the data.

We thank the editors for their assessment. We are excited to present a revised version of the manuscript with improved data presentation and an improved discussion addressing the reviewer’s concerns.

**Public Reviews:**

**Reviewer #1 (Public Review):**
In the submitted manuscript, Port et al. investigated the host and viral factors influencing the airborne transmission of SARS-CoV-2 Alpha and Delta variants of concern (VOC) using a Syrian hamster model. The authors analyzed the viral load profiles of the animal respiratory tracts and air samples from cages by quantifying gRNA, sgRNA, and infectious virus titers. They also assessed the breathing patterns, exhaled aerosol aerodynamic profile, and size distribution of airborne particles after SARS-CoV-2 Alpha and Delta infections. The data showed that male sex was associated with increased viral replication and virus shedding in the air. The relationship between co-infection with VOCs and the exposure pattern/timeframe was also tested. This study appears to be an expansion of a previous report (Port et al., 2022, Nature Microbiology). The experimental designs were rigorous, and the data were solid. These results will contribute to the understanding of the roles of host and virus factors in the airborne transmission of SARS-CoV-2 VOCs.
**Reviewer #2 (Public Review):**
This manuscript by Port and colleagues describes rigorous experiments that provide a wealth of virologic, respiratory physiology, and particle aerodynamic data pertaining to aerosol transmission of SARS-CoV-2 between infected Syrian hamsters. The data is particularly significant because infection is compared between alpha and delta variants, and because viral load is assessed via numerous assays (gRNA, sgRNA, TCID) and in tissues as well as the ambient environment of the cage. The paper will be of interest to a broad range of scientists including infectious diseases physicians, virologists, immunologists and potentially epidemiologists. The strength of evidence is relatively high but limited by unclear presentation in certain parts of the paper.Important conclusions are that infectious virus is only detectable in air samples during a narrow window of time relative to tissue samples, that airway constriction increases dynamically over time during infection limiting production of fine aerosol droplets, that variants do not appear to exclude one another during simultaneous exposures and that exposures to virus via the aerosol route lead to lower viral loads relative to direct inoculation suggesting an exposure dose response relationship.While the paper is valuable, I found certain elements of the data presentation to be unclear and overly complex.
**Reviewer #1 (Recommendations For The Authors):**

We thank the reviewer for their comments and their attention to detail. We have taken the following steps to address their suggestions and concerns.

However, the following concerns need to be issued.1. Summary seems to be too simple, and some results are not clearly described in the summary.

We have edited the summary and hope to have addressed the concerns raised by providing more information. We think that the summary includes all relevant findings.

“It remains poorly understood how SARS-CoV-2 infection influences the physiological host factors important for aerosol transmission. We assessed breathing pattern, exhaled droplets, and infectious virus after infection with Alpha and Delta variants of concern (VOC) in the Syrian hamster. Both VOCs displayed a confined window of detectable airborne virus (24-48 h), shorter than compared to oropharyngeal swabs. The loss of airborne shedding was linked to airway constriction resulting in a decrease of fine aerosols (1-10µm) produced, which are suspected to be the major driver of airborne transmission. Male sex was associated with increased viral replication and virus shedding in the air. Next, we compared the transmission efficiency of both variants and found no significant differences. Transmission efficiency varied mostly among donors, 0-100% (including a superspreading event), and aerosol transmission over multiple chain links was representative of natural heterogeneity of exposure dose and downstream viral kinetics. Co-infection with VOCs only occurred when both viruses were shed by the same donor during an increased exposure timeframe (24-48 h). This highlights that assessment of host and virus factors resulting in a differential exhaled particle profile is critical for understanding airborne transmission.”

1. Aerosol transmission experiment should be described in Materials and Methods although it is cited as Reference 21#;

We have modified Line 433:

“Aerosol caging

Aerosol cages as described by Port et al. [2] were used for transmission experiments and air sampling as indicated. The aerosol transmission system consisted of plastic hamster boxes (Lab Products) connected by a plastic tube. The boxes were modified to accept a 7.62 cm (3') plastic sanitary fitting (McMaster-Carr), which enabled the length between the boxes to be changed. Airflow was generated with a vacuum pump (Vacuubrand) attached to the box housing the naïve animals and was controlled with a float-type meter/valve (McMaster-Carr).”

And Line 458: “During the first 5 days, hamsters were housed in modified aerosol cages (only one hamster box) hooked up to an air pump.”.

Especially, one superspreading event of Alpha VOC (donor animal) was observed in iteration A (Figure 4). What causes that event, experiment system?

Based on the observed variation in airborne shedding (of the cages from which this was directly measured), we believe that one plausible explanation for the super-spreading event was that the Alpha-infected donor shed considerably more virus during the exposure than other donors, and thus more readily infected the sentinels. That said, it is also conceivable that other factors such as hamster behavior (e.g., closeness to the cage outlet, sleeping) or variable sentinel susceptibility could affect the distribution of transmissions.

1. Same reference is repeatedly listed as Refs 2 and 21#.

Addressed. We thank the reviewer for their attention to detail. We have also removed reference 53, which was the same as 54.

1. Two forms of described time (hour and h) are used in the manuscript. Single form should be chosen.

This has been addressed.

1. Virus designation located in line 371 and line 583 is inconsistent, and it needs to be revised.

For consistency we have chosen this nomenclature for the viruses used: SARS-CoV-2 variant Alpha (B.1.1.7) (hCoV320 19/England/204820464/2020, EPI_ISL_683466) and variant Delta (B.1.617.2/) (hCoV-19/USA/KY-CDC-2-4242084/2021, EPI_ISL_1823618).

1. In Figure 5F, what time were lung and nasal turbinate tissues collected after virus infection?

This has been added to the legend. Day 5. Line 904.

1. Line 562-563, what is the coating antigen (spike protein, generated in-house)? purified or recombinant protein?

It is in-house purified recombinant protein. This has been added to the methods.

1. Line 575 and line 578: 10,000x is not standard description, and it should be revised.

Done.

**Reviewer #2 (Recommendations For The Authors):**

We thank the reviewer for their comments and suggestions to improve the manuscript, and hope we have addressed all concerns adequately.

• Direct interpretation of the linear regression slope in Figure 3 is challenging. Is the most relevant parameter for transmission known? Intuitively, it would be the absolute number of small droplets at a given timepoint rather than the slope and it would be easier to interpret if the data were reported in this fashion.

We decided to show a percentage of counts to normalize the data among animals, as we observed large inter-individual variation in counts. The reviewer is correct that it is most likely the number of particles that would be most relevant to transmission, though much (including the role of particle size) remains to be determined. We have added a sentence to the results which explains this in L157.

Therefore, we decided in this first analysis to utilize the slope measurement and not raw counts. The focus was on the slopes and how particle profiles were changing post inoculation. Because we have focused on percentages, it seems not appropriate to present particle counts within each diameter range because the analysis, model, and results are based on these percentages of particles.

Use of regression to compute slope is a useful measure because it uses data from all timepoints to estimate the regression line and, therefore, the % of particles on each day. We decided on these methods because efficiency is especially important in a study with a relatively small number of animals and slopes are also a good surrogate for how animal particle profiles are changing post-inoculation.

To assist with the interpretation:

1. We removed Figure 3C and D and replaced Figure 3B with individual line plots for all conditions to visualize the slopes. The figure legend was corrected to reflect these changes.

2. We replaced L169 onwards to read: (Figure 3B). Females had a steeper decline at an average rate of 2.2 per day after inoculation in the percent of 1-10 μm particles (and a steeper incline for <0.53 μm) when compared to males, while holding variant group constant. When we compared variant group while holding sex constant, we found that the Delta group had a steeper decline at an average rate of 5.6 per day in the percent of 1-10 μm particles (and a steeper incline for <0.53 μm); a similar trend, but not as steep, was observed for the Alpha group.

The estimated difference in slopes for Delta vs. controls and Alpha vs. controls in the percent of <0.53 μm particles was 5.4 (two-sided adjusted p = 0.0001) and 2.4 (two-sided adjusted p = 0.0874), respectively. The estimated difference in slopes for percent of 1-10 μm particles was not as pronounced, but similar trends were observed for Delta and Alpha. Additionally, a linear mixed model was considered and produced virtually the same results as the simpler analysis described above; the corresponding linear mixed model estimates were the same and standard errors were similar.

• Fig 4: what is "limit of quality" mentioned in the legend? Are these samples undetectable?

We have clarified this in the legend: “3.3 = limit of detection for RNA (<10 copies/rxn)”. If samples have below 10 copy numbers per reaction, they are determined to be below the limit of detection. The limit of detection is 10 copy number/rxn. All samples below 10 copies/rxn are taken to be negative and set = 10 copies/rxn, which equals 3.3. Log10 copies/mL oral swab.

• Fig 4C would be easier to process in graphical rather than tabular form. The meaning of the colors is unclear.

We agree with the reviewer that this is difficult to interpret, but we are uncertain if the same data in a tabular format would be easier to digest. We realized that the legend was misplaced and have added this back into the figure, which we hope clarifies the colors and the limit of detection.

• Figure 4D & E are uninterpretable. What do the pie charts represent?

We have remodeled this part of the figure to a schematic representation of the majority variant which transmitted for each individual sentinel, and have added a table (Table S1) which summarizes the exact sequencing results for the oral swabs. The reviewer is correct that it was difficult to interpret the pie charts, considering most values are either 0 or close to 100%. We hope this addresses the question. The legend states:

**Author response image 1. sa3fig1:** Airborne attack rate of Alpha and Delta SARS-CoV-2 variants. Donor animals (N = 7) were inoculated with either the Alpha or Delta variant with 103 TCID50 via the intranasal route and paired together randomly (1:1 ratio) in 7 attack rate scenarios (A-G). To each pair of donors, one day after inoculation, 4-5 sentinels were exposed for a duration of 4 h (i.e., h 24-28 post inoculation) in an aerosol transmission set-up at 200 cm distance. (**A**) Schematic figure of the transmission set-up. (**B**) Day 1 sgRNA detected in oral swabs taken from each donor after exposure ended. Individuals are depicted. Wilcoxon test, N = 7. Grey = Alpha, teal = Delta inoculated donors. (**C**) Respiratory shedding measured by viral load in oropharyngeal swabs; measured by sgRNA on day 2, 3, and 5 for each sentinel. Animals are grouped by scenario. Colors refer to legend below. 3.3 = limit of detection of RNA (<10 copies/rxn). (**D**) Schematic representation of majority variant for each sentinel as assessed by percentage of Alpha and Delta detected in oropharyngeal swabs taken at day 2 and day 5 post exposure by deep sequencing. Grey = Alpha, teal = Delta, white = no transmission.

• Fig S2G is uninterpretable. Please label and explain.

We have now included an explanations of the figure S2F. The figure is a graphic representation of the neutralization data depicted in Figure S2F. The spacing between grid lines is 1 unit of antigenic distance, corresponding to a twofold dilution of serum in the neutralization assay. The resulting antigenic distance depicted between Alpha and Delta is roughly a 4-fold difference in neutralization between homologous (e.g., Alpha sera with the Alpha virus vs. heterologous, Alpha sera with the Delta virus).

• I would consider emphasizing lines 220-225 in the summary and abstract. The important implication is that aerosol transmission is more representative of natural heterogeneity of exposure dose and downstream viral kinetics. This is an often-overlooked point.

We agree with the reviewer and have added this in Line 43.

• Fig 5: A cartoon similar to Fig 4A showing timing of sentinel exposure with number of animals would be helpful.

We have added this as a new panel A for Figure 5. See the redrafted Figure 5 below.

• For Fig 5E & F It would be helpful to use a statistical test to more formally assess whether proportion at exposure predicts proportion of variants in downstream sentinel infection.

This has been added as a new Figure 5 panel H and I, which we hope addresses the reviewer’s comment.

**Author response image 2. sa3fig2:** Airborne competitiveness of Alpha and Delta SARS-CoV-2 variants. (**A**) Schematic. Donor animals (N = 8) were inoculated with Alpha and Delta variant with 5 x 102 TCID50, respectively, via the intranasal route (1:1 ratio), and three groups of sentinels (Sentinels 1, 2, and 3) were exposed subsequently at a 16.5 cm distance. Animals were exposed at a 1:1 ratio; exposure occurred on day 1 (Donors = Sentinels 1) and day 2 (Sentinels = Sentinels). (**B**) Respiratory shedding measured by viral load in oropharyngeal swabs; measured by gRNA, sgRNA, and infectious titers on days 2 and day 5 post exposure. Bar-chart depicting median, 96% CI and individuals, N = 8, ordinary two-way ANOVA followed by Šídák's multiple comparisons test. (**C/D/E**) Corresponding gRNA, sgRNA, and infectious virus in lungs and nasal turbinates sampled five days post exposure. Bar-chart depicting median, 96% CI and individuals, N = 8, ordinary two-way ANOVA, followed by Šídák's multiple comparisons test. Dark orange = Donors, light orange = Sentinels 1, grey = Sentinels 2, dark grey = Sentinels 3, p-values indicated where significant. Dotted line = limit of quality. (**F**) Percentage of Alpha and Delta detected in oropharyngeal swabs taken at days 2 and day 5 post exposure for each individual donor and sentinel, determined by deep sequencing. Pie-charts depict individual animals. Grey = Alpha, teal = Delta. (**G**) Lung and nasal turbinate samples collected on day 5 post inoculation/exposure. (**H**) Summary of data of variant composition, violin plots depicting median and quantiles for each chain link (left) and for each set of samples collected (right). Shading indicates majority of variant (grey = Alpha, teal = Delta). (**I**) Correlation plot depicting Spearman r for each chain link (right, day 2 swab) and for each set of samples collected across all animals (left). Colors refer to legend on right. Abbreviations: TCID, Tissue Culture Infectious Dose.”

We have additionally added to the results section: L284: “Combined a trend, while not significant, was observed for increased replication of Delta after the first transmission event, but not after the second, and in the oropharyngeal cavity (swabs) as opposed to lungs (Figure 5H) (Donors compared to Sentinels 1: p = 0.0559; Donors compared to Sentinels 2: p = >0.9999; Kruskal Wallis test, followed by Dunn’s test). Swabs taken at 2 DPI/DPE did significantly predict variant patterns in swabs on 5 DPI/DPE (Spearman’s r = 0.623, p = 0.00436) and virus competition in the lower respiratory tract (Spearman’s r = 0.60, p = 0.00848). Oral swab samples taken on day 5 strongly correlate with both upper (Spearman’s r = 0.816, p = 0.00001) and lower respiratory tract tissue samples (Spearman’s r = 0.832, p = 0.00002) taken on the same day (Figure 5I).”

• Fig 1A: how are pfu/hour inferred? This is somewhat explained in the supplement, but I found the inclusion of model output as the first panel confusing and am still not 100% clear how this was done. Consider, explaining this in the body of the paper.

We have added a more detailed explanation of the PFU/h inference to the main text: The motivation for the model was to link more readily measurable quantities such as RNA measured in oral swabs to the quantity of greatest interest for transmission (infectious virus per unit time in the air). To do this, we jointly infer the kinetics of shed airborne virus and parameters relating observable quantities (infected sentinels, plaques from purified air sample filters) to the actual longitudinal shedding. The inferential model uses mechanistic descriptions of deposition of infectious virus into the air, uptake from the air, and loss of infectious virus in the environment to extract estimates of the key kinetic parameters, as well as the resultant airborne shedding, for each animal.

We have added this information to L106 in the results and hope this clarifies the rationale and execution of the model.

More minor points:• Line 292: "poor proxy" seems too strong as peak levels of viral RNA correlate with positive airway cultures. It might be more accurate to say that high levels of viral RNA during early infection only somewhat correlate with positive airway cultures.

We have rephrased this to clarify that while peak RNA viral loads are predictive of positive cultures, measuring RNA, especially early during infection and only once, may not be sufficient to infer the magnitude or time-dependence of infectious virus shedding into the air. See Line 308: “We found that swab viral load measurements are a valuable but imperfect proxy for the magnitude and timing of airborne shedding. Crucially, there is a period early in infection (around 24 h post-infection in inoculated hamsters) when oral swabs show high infectious virus titers, but air samples show low or undetectable levels of virus. Viral shedding should not be treated as a single quantity that rises and falls synchronously throughout the host; spatial models of infection may be required to identify the best correlates of airborne infectiousness [32]. Attempts to quantify an individual’s airborne infectiousness from swab measurements should thus be interpreted with caution, and these spatiotemporal factors should be considered carefully.”

• Line 352: Re is dependent on time of an outbreak (population immunity) and cannot be specified for a given variant as it depends on multiple other variables

We agree that the current phrasing here could be interpreted to suggest, incorrectly, that Re is an intrinsic property of a variant. We have deleted that language and reworded the section to emphasize that the critical question is heterogeneity in transmission, not mean reproduction number. Line 348: “Moreover, at the time of emergence of Delta, a large part of the human population was either previously exposed to and/or vaccinated against SARS-CoV-2; that underlying host immune landscape also affects the relative fitness of variants. Our naïve animal model does not capture the high prevalence of pre-existing immunity present in the human population and may therefore be less relevant for studying overall variant fitness in the current epidemiological context. Analyses of the cross-neutralization between Alpha and Delta suggest subtly different antigenic profiles [35], and Delta’s faster kinetics in humans may have also helped it cause more reinfections and “breakthrough” infections [36].

Our two transmission experiments yielded different outcomes. When sentinel hamsters were sequentially exposed, first to Alpha and then to Delta, generally no dual infections—both variants detectable—were observed. In contrast, when we exposed hamsters simultaneously to one donor infected with Alpha and another infected with Delta, we were able to detect mixed-variant virus populations in sentinels in one of the cages (Cage F, see Appendix figures S1, S2). The fact that we saw both single-lineage and multi-lineage transmission events suggests that virus population bottlenecks at the point of transmission do indeed depend on exposure mode and duration, as well as donor host shedding. Notably, our analysis suggests that the Alpha-Delta co-infections observed in the Cage F sentinels could be due to that being the one cage in which both the Alpha and the Delta donor shed substantially over the course of the exposure (Appendix figures S2, S3). Mixed variant infections were not retained equally, and the relative variant frequencies differed between investigated compartments of the respiratory tract, suggesting roles for randomness or host-and-tissue specific differences in virus fitness.

A combination of host, environmental and virus parameters, many of which vary through time, play a role in virus transmission. These include virus phenotype, shedding in air, individual variability and sex differences, changes in breathing patterns, and droplet size distributions. Alongside recognized social and environmental factors, these host and viral parameters might help explain why the epidemiology of SARS-CoV-2 exhibits classic features of over-dispersed transmission [37]. Namely, SARS-CoV-2 circulates continuously in the human population, but many transmission chains are self-limiting, while rarer superspreading events account for a substantial fraction of the virus’s total transmission. Heterogeneity in the respiratory viral loads is high and some infected humans release tens to thousands of SARS-CoV-2 virions/min [38, 39]. Our findings recapitulate this in an animal model and provide further insights into mechanisms underlying successful transmission events. Quantitative assessment of virus and host parameters responsible for the size, duration and infectivity of exhaled aerosols may be critical to advance our understanding of factors governing the efficiency and heterogeneity of transmission for SARS-CoV-2, and potentially other respiratory viruses. In turn, these insights may lay the foundation for interventions targeting individuals and settings with high risk of superspreading, to achieve efficient control of virus transmission [40].”

• The limitation section should mention that this animal model does not capture the large prevalence of pre-existing immunity at present in the population and may therefore be less relevant in the current epidemiologic context.

We agree and have added this more clearly, see response above.

• Limitation: it is unclear if airway and droplet dynamics in the hamster model are representative of humans.

We have added the following sentence: Line 331: “It remains to be determined how well airway and particle size distribution dynamics in Syrian hamsters model those in humans.”

• The mathematical model is termed semi-mechanistic but I think this is not accurate as the model appears to have no mechanistic assumptions.

We describe the model as semi-mechanistic because it uses mechanistic descriptions of the shedding and uptake process (as described above), incorporating factors including respiration rate and environmental loss, and makes the mechanistic assumption that measurable swab and airborne shedding all stem from a shared within-host infection process that produces exponential growth of virus up to a peak, followed by exponential decay. The model is only semi-mechanistic, however, as we do not attempt a full model of within-host viral replication and shedding (e.g. a target-cell limited virus kinetics model).